

# Digital Elevation Models and Orthomosaics of 1989 Aerial Imagery of the Western Antarctic Peninsula and Surrounding Islands between 66-68°S

Vijaya Kumar Thota[1], Thorsten Seehaus[1], Friedrich Knuth[2,3], Amaury Dehecq[4], Christian Salewski[5], and Matthias Braun[1]

[1]Institut für Geographie, Friedrich-Alexander-Universität Erlangen-Nürnberg, Erlangen, Germany
[2]Laboratory of Hydraulics, Hydrology and Glaciology (VAW), ETH Zurich, Zurich, Switzerland
[3]Swiss Federal Institute for Forest, Snow and Landscape Research (WSL), bâtiment ALPOLE, Sion, Switzerland
[4]Univ. Grenoble Alpes, IRD, CNRS, INRAE, Grenoble INP, IGE, 38000 Grenoble, France
[5]Archiv für deutsche Polarforschung, Alfred-Wegener-Institut, Bremerhaven, Germany

**Correspondence:** Vijaya Kumar Thota (vijaya.kumar.thota@fau.de)

**Abstract.** We present a unique, timestamped, high-resolution Digital Elevation Model (DEM) and orthomosaic dataset, derived from aerial imagery that covers about 12000 km$^2$ area on the western Antarctic Peninsula and surrounding islands between 66-68°S. We used a film-based aerial image archive from 1989 acquired by the Institut für Angewandte Geodäsie (IfAG), and is kept in the Archive for German Polar Research at the Alfred Wegener Institute, Germany, to generate the historical DEMs and orthoimages. The reference elevation model of Antarctica (REMA) mosaic is used as a reference DEM to co-register our historical product on stable ground. We evaluated the vertical accuracy of the derived IfAG DEM with independent surface elevation data from ICESat-2 from the summer months of 2020 and 2021. Our historical DEMs have vertical accuracies better than 6 m and 8 m with respect to modern elevation data, REMA, and ICESat-2, respectively. The late 20th century DEM and orthomosaic are very valuable observations in a data sparse region, and this dataset will help to quantify historical ice volume changes and inform geodetic mass balance estimates. The dataset is publicly available at https://doi.org/10.5281/zenodo.16836526 (Thota et al., 2025) and the results presented in this paper are based on version 1.1 of the dataset.

## 1 Introduction

Monitoring of glaciers dates back to 1830 (Clarke, 1987), when researchers used various instruments, starting from a boulder (fixed marker), stakes, theodolites, and photographs to extract glacier length, area, and volume (mass) changes (Zemp et al., 2015; Oerlemans, 2005). Over time, advancements in remote sensing and imaging technology have expanded glacier monitoring capabilities, thus establishing regional-level monitoring of glaciers for length, area, and volume changes since the early twenty-first century (Berthier et al., 2016; Braun et al., 2019; Sommer et al., 2020; Hugonnet et al., 2021; Seehaus et al., 2023; The GlaMBIE Team, 2025). The World Glacier Monitoring Service (WGMS) standardizes and provides the mass balance of glaciers available through field measurements (Zemp et al., 2015). However, out of approximately 200000 glaciers worldwide, only 37, mostly in accessible regions like the Alps, have continuous mass balance records dating back to the 20th century



(Zemp et al., 2015). Although more glaciers have at least one observation prior to 2000, such historical data remain sparse for the Antarctic Peninsula (AP), a key region of rapid warming in recent decades (Turner et al., 2016; Oliva et al., 2017; Dussaillant et al., 2025).

The archives of over 30,000 aerial photographs from the Antarctic Peninsula acquired since the 1940s are the only direct observations available over the last century to reconstruct past glacier surface elevations (Fox and Cziferszky, 2008). In 1956-57, the Falkland Islands and Dependencies Aerial Survey Expedition (FIDASE) undertook extensive aerial mapping surveys throughout the AP. More than 12,000 images were taken along the 26.000 km of ground track (Dodds, 1996; Mott and Wiggins, 1965). Moreover, wide parts of the AP were covered by U.S. aerial surveys in the 1960s. Trimetrogon imagery (a camera system consisting of three cameras, one pointing down, the other two pointed to either side of the flight path at a 30° depression angle) was acquired during these surveys(Dahle et al., 2024). In 1989, the Institut für Angewandte Geodäsie (IfAG) carried out survey flights along the western coast of the AP north of Marguerite Bay near Adelaide Island using the German Research Vessel Polarstern and its helicopters. Large glacierized areas were covered by overlapping vertical photographs. Long-term mass balance analysis from these archives exists for a limited number of glaciers on the Antarctic Peninsula and surrounding Islands. For instance, Kunz et al. (2012) combined various U.S. and U.K. airborne and ASTER spaceborne stereo imagery to estimate glacier surface elevation changes for 12 glaciers on the western AP between 1948 and 2010 (primarily 1960s - 2005). They revealed a mean near-frontal surface lowering rate of 0.28 +/- 0.03 m/a since the mid-1960s and an increased lowering since the 1990s. Fieber et al. (2016) carried out a case study at Lindblad Cove, north-western AP. They obtained surface elevation change data by combining historical aerial imagery and WorldView-2 satellite stereo-imagery for the period 1957-2014. A total positive mass balance between 0.6 and 5.8 m w.e. was observed throughout the study period. In a follow-up analysis, Fieber et al. (2018) analyzed the surface elevation changes of 16 individual glaciers, grouped at 4 locations on the AP and surrounding islands, between 1956 and 2014. So far, the IfAG data has been analyzed only for the San Martín region (McGlary & Northeast glaciers) (Wrobel et al., 2000) and for carrying out a case study at Moider Glacier by Fox and Cziferszky (2008). The results from the different long-term measurements of glacier surface elevation changes indicate a heterogeneous change pattern and suggest that upscaling of the sparse measurements on regional scales can be significantly biased by the limited availability of data.

Here, we present a unique, timestamped, high-resolution Digital Elevation Model (DEM) and orthomosaic dataset of aerial imagery that covers about 12000 km$^2$ on the western Antarctic Peninsula and surrounding islands between 66-68°S. The data has been generated with detailed analysis of around 1200 film-based aerial images from 1989 acquired by the Institut für Angewandte Geodäsie (IfAG), Germany, using Multiview Structure from Motion (MV-SfM) methods. The DEMs have been co-registered and evaluated against external elevation data such as Reference Elevation Model of Antarctica (REMA) and ICESat-2 (Howat et al., 2019).





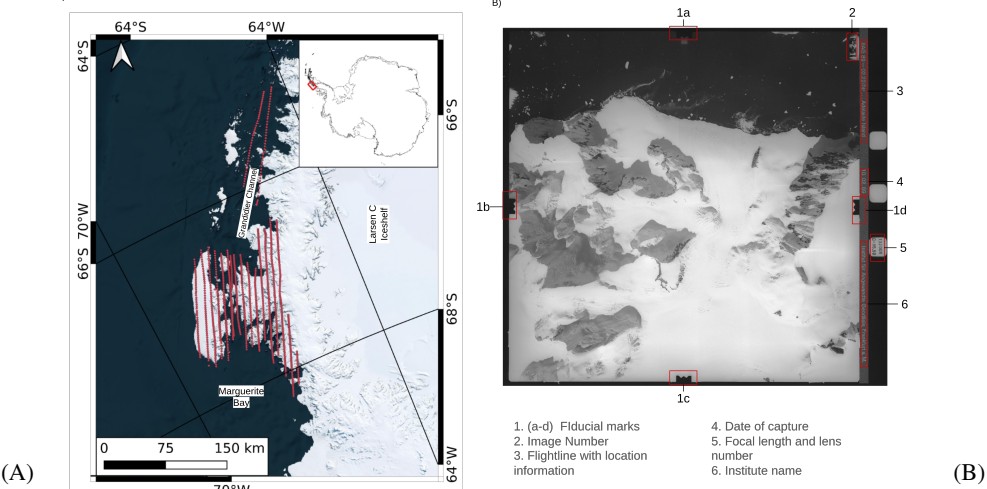

**Figure 1.** (A) IfAG aerial imagery archive, with red dots representing initial camera locations digitized from survey index map, (B) Scanned IfAG image with metadata information highlighted in red (1-6)

## 2 Data

### 2.1 IfAG aerial imagery archive

The archive consists of approximately 2000 aerial images, acquired during a photogrammetric survey by the Institut für Ange-wandte Geodäsie (IfAG), Frankfurt am Main, Germany. The survey covered the areas on the western Antarctic Peninsula near Adelaide Island and the Grandidier Channel between 06 February 1989 and 20 February 1989. Aerial photographs were taken using a Zeiss RMK A 8.5/23 camera with Agfa Aviphot Pan 200 aerial film at multiple image scales (1:70000, 1:30000, 1:15000, 1:10000, 1:5000) at different locations. Most of the images are captured with more than 60 % forward overlap and are flown at an average elevation of 5895 m, giving a nominal photoscale of 1:70000. The film positives used here are pre-served in the Archive for German Polar Research (Archive für deutsche Polarforschung - AdP) at the Alfred Wegener Institute (AWI) in Bremerhaven, Germany. There, they form part of the aerial image archive of the Federal Agency for Cartography and Geodesy (Bundesamt für Kartographie und Geodäsie - BKG), which is part of the Federal Ministry of the Interior and the legal successor to the IfAG. This collection comprises a total of more than 20,000 images and was donated to the AdP/AWI in 2017 by the BKG together with the necessary rights of use. The films were digitized using a Leica DSW 700 scanner by GTA Geoservice GmbH, Neubrandenburg, Germany. Image positives that are 23 cm x 23 cm are scanned at an average scanning resolution of 12.5 micrometers and provided in 8-bit radiometric resolution, resulting in digital images of size 20232 x 18829 ($\sim$380.948 x $10^6$) pixels (Figure 1). The average ground sampling distance (GSD) of these images corresponds to 0.875 m. To our knowledge, no scientific publication has extensively used this unique archive to study glacier elevation changes to date.



For this study, we selected images from IfAG dataset that were acquired at a uniform scale of 1:70000. These images were selected due to their consistent coverage and suitability for generating historical DEMs across the study area. Higher-resolution images targeting specific glacier front locations (taken at image scales of 1:30000, 1:15000, 1:10000, and 1:5000) are excluded from the analysis, as identifying stable areas for co-registering historical DEMs to modern reference datasets proved challenging at the individual glacier scale.

### 2.1.1 Auxiliary data

We used the Reference Elevation Model of Antarctica (REMA) mosaic as a reference DEM to extract stable (or static) ground elevation for co-registering our historical DEMs derived from the aerial imagery. The REMA mosaic is compiled from multiple REMA strips that are generated using very high resolution (0.32 to 0.5 m) WorldView-1,2,3 and GeoEye-1 satellite imagery through Surface Extraction from TIN-based Searchspace Minimization (SETSM) software (Howat et al., 2019). The mosaic is created to provide a more consistent and complete DEM product with blending and feathering of strip DEMs to avoid edge artifacts. REMA mosaic tiles are co-registered with satellite altimetry from Cryosat-2 and Icesat and have absolute vertical uncertainties of less than 1 m. Given that REMA is the only high-resolution DEM available with high accuracy, it can be used as an appropriate reference DEM for processing the IfAG data. We used ICESat-2 ATL06 L3A Land Ice Height data from the summer months of 2020 and 2021 to evaluate the vertical accuracy of the derived IfAG DEMs on ice-free areas. Glacier outlines and rock outcrops are taken from the Antarctic Digital Database and Silva et al. (2020).

## 3 Methods

Photogrammetric analysis of our approach is adopted from the Historical Structure from Motion (HSfM) workflow by Knuth et al. (2023) and primarily involves three steps (1) Preparing the scanned imagery for the photogrammetric processing (2) Applying MV-SfM to scanned imagery with estimated camera pose and focal lengths to generate DEMs (3) Multi-stage co-registration of coarsely geolocated DEMs to reference terrain. A detailed description of our approach is provided in the following sections (Figure 2).

### 3.1 Camera extrinsic estimation

To determine the planimetric information (longitude and latitude) of image centers, we manually estimated their locations using the survey index map from the Institut für Angewandte Geodäsie (IfAG) at a scale of 1:500,000, archived at AWI. The map was digitized and georeferenced to establish the initial camera locations. Planimetric coordinates of the image centers are extracted with respect to the WGS84 datum. The approximate elevation of the camera flown at a scale of 1:70000 is 5985 m above ground. We sample the terrain elevation relative to the WGS84 ellipsoid at each initial horizontal location of the camera positions from the REMA mosaic. Therefore, an initial flying height above the ellipsoid (geodetic height) is incorporated as a third dimension into the initial 3D coordinates of the cameras. We input these 3D coordinates with an accuracy estimate of 1000 m to Agisoft Metashape, along with initial Yaw, Pitch, Roll of 0°,0°,0°, with accuracies of 180°,10°,10°, respectively.

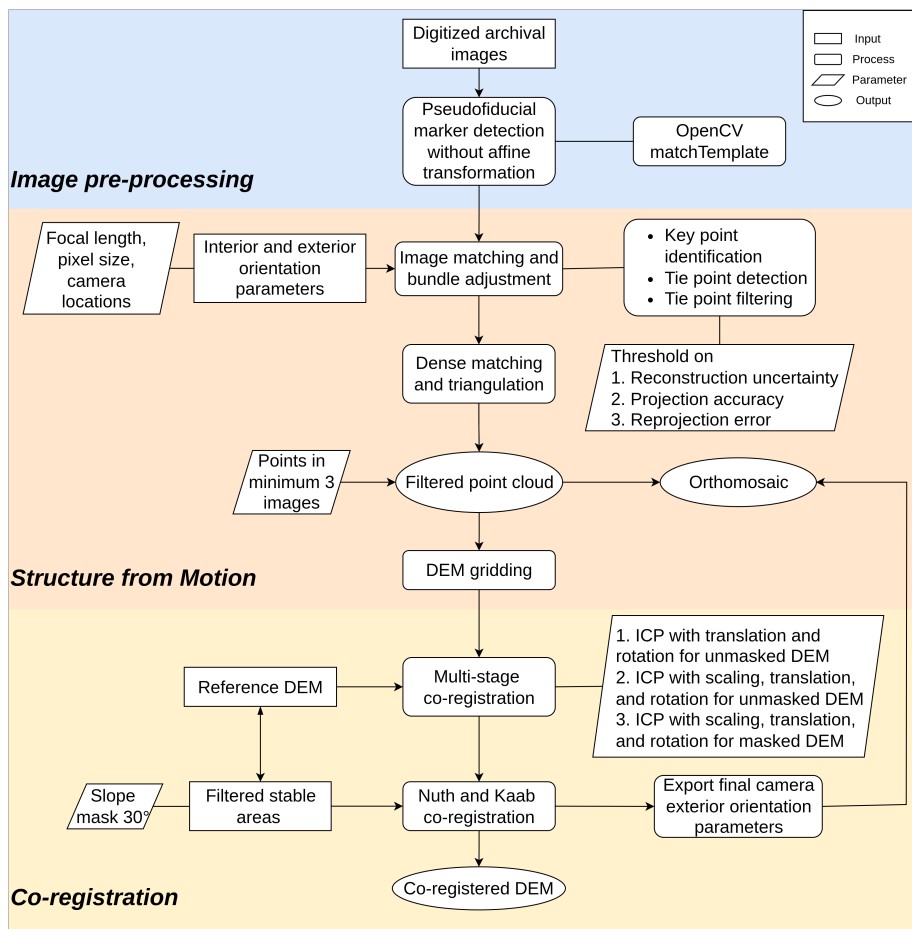

**Figure 2.** Historical Structure from Motion workflow, adapted from Knuth et al. (2023), ICP = Iterative Closest Point algorithm

## 3.2 Camera intrinsics estimation

### 3.2.1 Fiducial marker detection

The IfAG data comes with pseudofiducial markers (similar to fiducials, but with no clear marker center as opposed to a cross, for example) to secure the interior orientation of the camera (as shown in Figure 1). These markers serve as reference points for determining the camera's internal geometry at the time of image acquisition. We employed the template matching approach described in Pseudofiducial Marker Detection Without Affine Transformation (Knuth et al., 2021b) to automatically detect the locations of each of the four fiducial markers in around 1,200 images. The detection algorithm was configured to identify marker positions within a threshold of 40 pixels from the median position of all matches for each fiducial marker across each flightline.





The detection results were as follows, for 1193 images, 48.19 % of images had all four fiducial markers detected, 29.50 % of three fiducial markers detected; 15.34 % of images had two markers detected; 5.95 % of images had only one marker detected; and 1 % of images had no detectable fiducial markers. The principal point of each image was calculated as the centroid of the detected fiducial markers. A minimum of two fiducial points is required to determine the principal point. We programmatically estimated the principal point in 93 % of the images. Remaining images with fewer than two fiducials detected, predominantly covering ocean areas in flightlines extending outside glaciated areas, were excluded from further analysis. Subsequently, each image was cropped around the principal point to a fixed square dimension corresponding to the metric frame of the Zeiss camera, which has a physical dimension of 226 mm, equivalent to 18,080 pixels (McNabb et al., 2020).

### 3.2.2 Camera model estimation

The camera calibration reports of the IfAG survey were not found in the AWI archive and were most likely lost before the imagery was transferred to AWI. We therefore estimated unavailable camera intrinsics, i.e, radial and tangential lens distortion parameters, using self-calibration during bundle adjustment performed in Metashape. To estimate the distortion parameters of the camera used for the survey, we chose Pourquoi Pas Island (PPI, see Figure 3) as calibration site. PPI is situated 50 km east of Rothera station and contains representative terrain types for the entire Antarctic Peninsula (Cziferszky et al., 2010). Moreover, the higher image quality, featuring cloud-free imagery, strong contrast of IfAG data here made the PPI area of interest suitable for optimizing camera parameters. We generated a DEM from 27 images from PPI, with initial estimated camera positions and focal length of 85.5 mm, iteratively minimizing the residual error with respect to stable area in the REMA strip DEMs from 2019. Using Metashape's default Brown-Conrady lens distortion model (Duane, 1971), we solved for a subset of intrinsic parameters during bundle adjustment, i.e, principal point coordinates (Cx, Cy), as well as radial (K1, K2, K3) and tangential (P1, P2) lens distortion coefficients. These final intrinsic camera model parameters derived from PPI were then held fixed and used to process the entire dataset.

### 3.3 DEM generation

We processed approximately 550 images from 12 flightlines photogrammetrically in Agisoft Metashape version 2.1.1 in 8 subsets (Figure 4). The subsets are selected to optimize computational efficiency while encompassing well-distributed stable areas for co-registration, ensuring high absolute accuracy. We excluded images from the western part of Adelaide Island, due to the lack of stable areas and insufficient image features, and from north of Adelaide Island near Grandidier Channel, where images predominantly cover water pixels (Figure 1). The Structure from Motion (SfM) workflow is run on a computer with an NVIDIA RTX 500 Ada Generation GPU (32251 MB, 100 compute units, 2550MHz) and a CPU AMD Ryzen Threadripper PRO 5955WX 16-Cores 128 GB RAM. All point cloud generation parameters are detailed in Table 1.

Tie points were generated at the native resolution of the imagery, facilitating precise and independent alignment of each subset. We set thresholds on quality parameters to reduce the number of incorrect tie points (Over et al., 2021). Initially, we removed all tie points that have a reprojection accuracy of more than 10, which is equivalent to a camera base to height ratio of



**Table 1.** Parameters used in Agisoft Metashape to generate the point cloud

| Parameter | Value |
|---|---|
| Focal Length | 85.5 mm |
| Pixel Size | 0.0125 mm |
| Subsets | 8 Numbers — 6 belong to the Mainland: North1, North2, Arrowsmith1, Arrowsmith2, South1, South2. One subset on Adelaide Island, one subset on Pourquoi Pas Island |
| Alignment Parameters | High quality, 10,000 Tie points, 100,000 Key points |
| Reference Accuracy | Positional ±1000 m, Yaw 180°, Roll and Pitch ±10° |
| Point Cloud Parameters | Medium quality, aggressive filtering |
| Gridding Resolution | 10 m × 10 m |

**Table 2.** Point cloud quality vs. uncertainty and coverage metrics for PPI subset

| Point Cloud Quality | Image Scale | Points | Filtered Points | Mean Reproj. Error | Elev. Diff. Median (m) | Elev. Diff. NMAD (m) | Coverage (%) | DEM Res (m) |
|---|---|---|---|---|---|---|---|---|
| Ultrahigh | 1 | 335344271 | 105882921 | 0.602 | -0.03 | 1.69 | 20.60 | 5 |
| Medium | 1/4 | 48495836 | 28476349 | 0.602 | 0.06 | 1.93 | 44.58 | 10 |
| Lowest | 1/16 | 4395902 | 3025030 | 0.602 | 0.31 | 5.72 | 71.08 | 40 |

2.3 (parallax angle of 23°). This removes all points that have a poor viewing angle, which may lead to weak 3D reconstruction and increased uncertainty in depth estimation.

We filtered out tie points with poorly localized projections due to their larger size by applying a projection accuracy threshold of 5. This metric represents the average image scale of a feature across overlapping images. Then, we reduced the reprojection errors of all subsets to less than 0.5 pixels. The filtered tie points are then used to generate a dense point cloud for subsets at medium quality, i.e., at a scale sixteen times less than the original scale of the images. We selected a medium accuracy in "depth map generation" as a compromise between the required accuracy and coverage (Table 2). To increase the robustness of the generated DEMs, we further excluded the points that were found in fewer than three scenes. We then generated a DEM by gridding the filtered point cloud at 10 m posting in the Antarctic Polar Stereographic (EPSG code 3031) coordinate system. This resolution corresponds to approximately three times the effective GSD of the input images processed at medium quality (~3.5 m), thereby minimizing interpolation artifacts and aligning with the resolution of the 10 m REMA DEM to avoid additional resampling.





## 3.4 Co-registration

We used the Ames Stereo Pipeline (ASP v3.4)'s *pc_align.py* tool that is embedded in HSfM for multistage co-registration of the generated raw DEMs, based on the Iterative Closest Point (ICP) algorithm (Beyer et al., 2018; Knuth et al., 2023; Shean et al., 2016). At each stage of ICP co-registration, a 12-parameter transformation matrix is calculated according to the expected offset and differences between the reference and raw DEM. In the first step, the rigid body transformation of the point-to-plane algorithm with translation and rotation is applied to the whole raw DEM with respect to the reference DEM (REMA mosaic). In the second step, in addition to translation and rotation, scaling is also added to further converge the offsets to the reference DEM. It is achieved by a similarity-point-to-plane algorithm. The point-to-plane algorithm is more robust to outliers than the point-to-point algorithm and thus converges faster (Li et al., 2020; Shean et al., 2016). In the last stage of ICP co-registration, refined the alignment by appling translation, rotation, and scaling corrections to stable areas only. Ice-free areas are taken from Silva et al. (2020) and the Antarctic Digital Database (ADD) rock outcrop mask. Stable areas for co-registration are determined by: 1. Filtering these areas to exclude slopes greater than 30° and minimize steep terrain-induced errors. 2. Manually removing blunders in areas where feature matching failed by cross-referencing with the Landsat Image Mosaic of Antarctica (LIMA) and orthoimages (Figure 3). We set the default expected offset values at each stage of the three-step ICP co-registration procedure to 2500, 500, and 100 m, based on visual inspection of multiple IfAG DEMs. After ICP co-registration, we applied the Nuth and Kääb (2011) algorithm over stable areas to further achieve subpixel alignment of the DEMs.

Finally, we mosaicked the 6 subsets that belong to the mainland Antarctic Peninsula into one product using the average value of the overlapping area to create a mosaic DEM. We provide DEMs that belong to two islands, i.e., Adelaide Island, PPI, as two separate files. We set the minimum elevation value to 0 and reclassified DEMs to eliminate negative values. We also provide a "bad" pixel mask raster for the mainland DEM, where our DEM is affected due to clouds or artefacts in the reference DEM. We generated this mask by excluding outlier elevation differences with respect to REMA mosaic (values exceeding 4 standard deviations from the mean) within glacier-covered areas, calculated across a binned elevation raster. We suggest using the DEM data where the bad pixel mask raster value is 1.

## 3.5 Orthoimage generation

The IfAG camera extrinsics are updated by applying the transformation matrix obtained from ICP co-registration and 3D shift vector from Nuth and Kääb (2011) co-registration of IfAG DEMs, and orthomosaics are generated using Metashape at the original resolution. We use Metashape's void-filling option of the DEMs to generate orthoimages without voids. We provide orthomosaics of 8 subsets in 8 TIFF files.

## 3.6 Uncertainty estimation

Uncertainties are estimated with two independent datasets, one with the reference DEM used to generate the historical DEMs, REMA mosaic, and the second with the ICESat-2 data. Uncertainties in the DEMs with respect to REMA are calculated following the approach described by Seehaus et al. (2019). First, elevation offsets (dh) are extracted in ice-free areas, which are
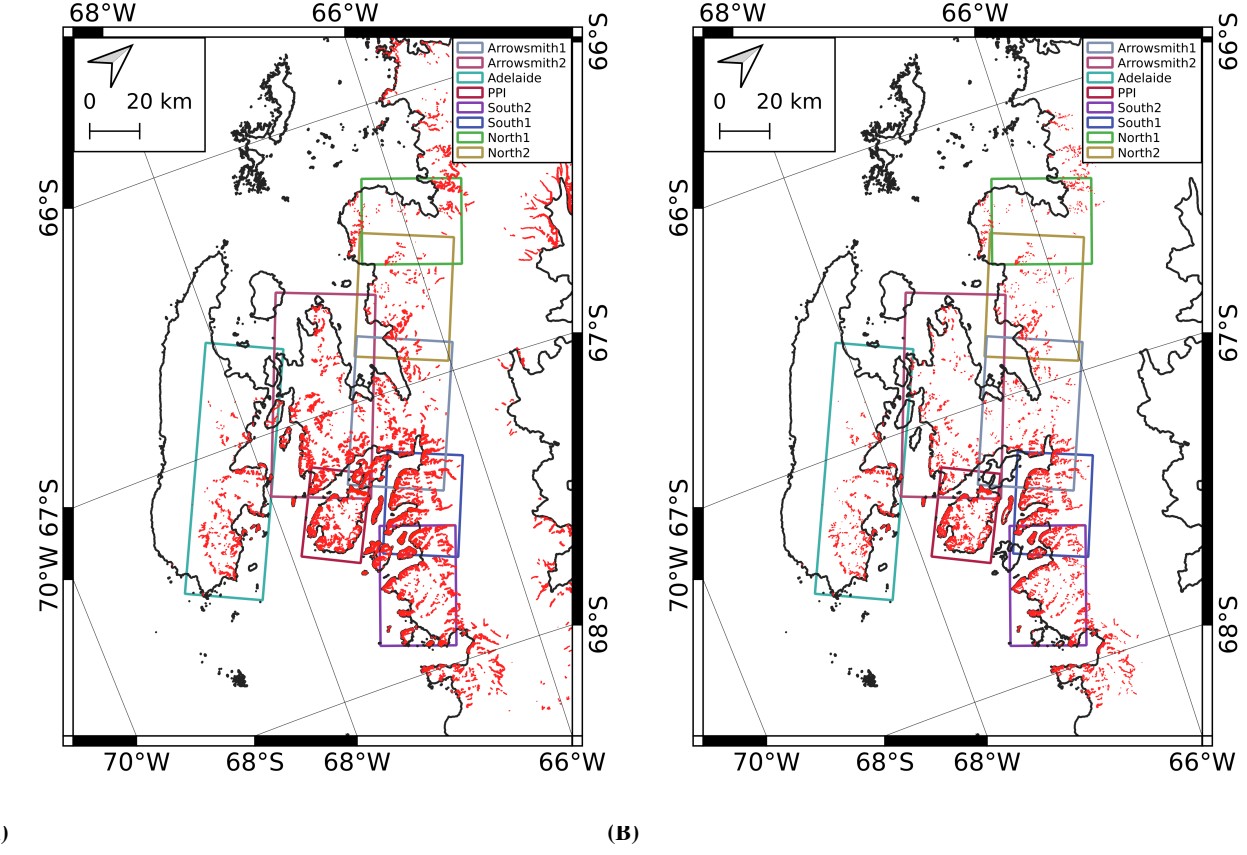

**(A)**                                                                    **(B)**

**Figure 3.** (A) Ice-free areas taken from Silva et al. (2020), (B) Stable areas used for co-registration in our study - slopes < 30° and manually filtered cross-referencing with LIMA and orthoimages, Background- High resolution vector polygons of the Antarctic coastline V7.8 (Gerrish et al., 2023)

then filtered for outliers using 2-98 percentiles of the data. These dh values are binned in 5 degrees slope intervals. Remaining outliers are filtered by applying a 3 times Normalized Median Absolute Deviation (NMAD) filter in each slope bin. The uncertainty of IfAG DEMs is estimated with respect to ICESat-2 data by evaluating the distribution of elevation differences between the two on stable areas. We first removed the gross outliers in the offsets by retaining data of less than 50 m offsets, accounting for errors caused by the cloud cover. Subsequently, we applied a 2–98 percentile filter to the entire dataset to exclude extreme values to suppress the impact of processing artifacts. Finally, dh values outside of 3 NMADs were removed across the entire dataset to ensure robust outlier elimination.



## 4 Results and Discussion

Our processing of the 1989 IfAG aerial imagery archive resulted in three Digital Elevation Models (DEMs) and six orthomosaics covering the mainland Antarctic Peninsula (comprising six subsets: North1, North2, Arrowsmith1, Arrowsmith2, South1, South2) and two for Pourquoi Pas Island (PPI) and Adelaide Island (Figure 4,Figure 5). In the following sections, we discuss key findings related to the dataset's coverage and image quality, adjustments to camera orientation (exterior and interior), and the vertical accuracy of the DEMs compared to reference datasets (REMA and ICESat-2) and other historical DEMs.

### 4.1 Image Quality and Coverage

We estimated the Shannon entropy of each of the images to analyze the image quality and examine any correlation to the coverage. We used this to measure the texture of the input images. Shannon entropy measures the variability of the data based on the probability of occurrence and is described in Equation 1:

$$H(X) = -\sum_{i=1}^{n} p(x_i) \log_2 p(x_i) \tag{1}$$

where $H(X)$ is the entropy of the random variable $X$ and represents the average level of information or uncertainty inherent in the possible outcomes, $p(x_i)$ is the probability of the $i$-th outcome $x_i$, $\log_2$ denotes the logarithm base 2, and $n$ is the total number of possible outcomes. The Shannon entropy value ranges from 0 to $log_x n$, where $n$ is the number of bins (for an 8-bit image is 256) and $x$ is the base. We here used a base of 2, therefore, the range is 0-8 for all of our images. For the IfAG archive, the Shannon entropy value ranged from 4.3 to 7.5, with an average of 6.76 for all selected images (Figure 4). No scanning-related artefacts are observed in the processed DEMs. Terrain shadows are visible in some places in the South1 subset and may be the cause of 7.18 (high) entropy value in these areas. In Table 3, we summarize the percentage area coverage of two land classes in our study area 1. Glacier 2. Ice-free areas. We masked our DEMs using the respective layers from Silva et al. (2020). We estimate area coverage percentages by dividing the number of valid pixels in a class by the total pixels of the class. Overall coverage (Glacier + Ice-free Areas) of all DEMs ranges between 20 and 45 %, with on-glacier coverages spanning 20-42 %. Notably, DEMs for PPI and North1, South1 show higher coverage, which corresponds to the relatively higher average entropy of their source images (Figure 4,Figure 5).

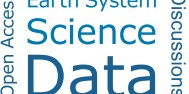

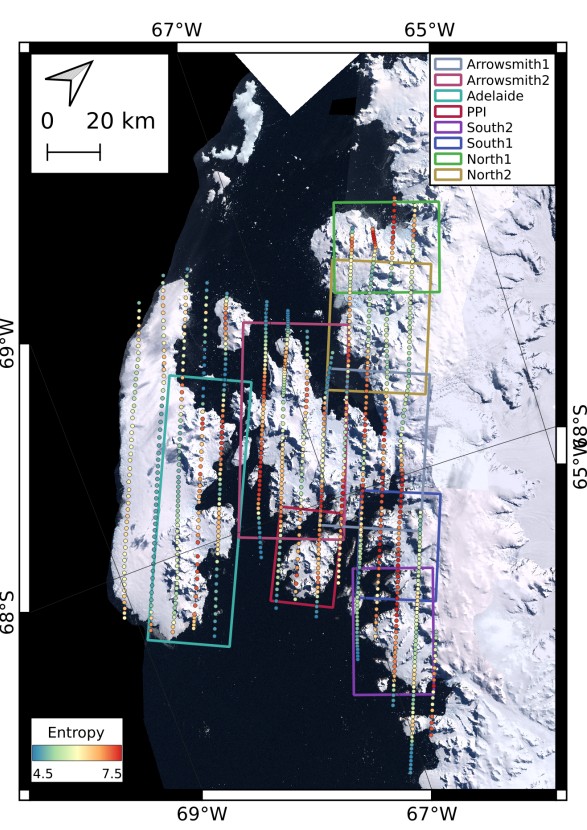

**Figure 4.** Subsets considered in this study, with rectangles representing subsets, 8 Nos — 6 belong to the Mainland: North1, North2, Arrowsmith1, Arrowsmith2, South1, South2. One subset on Adelaide Island, one subset on Pourquoi Pas Island, and dots representing image entropy (red-high, blue-low), with a background LIMA

**Figure 5.** (A) Derived DEMs as hillshade, with a background (B) Orthomosaics, with a background High resolution vector polygons of the Antarctic coastline V7.8 (Gerrish et al., 2023) (C) Decadal elevation change between IfAG DEM from 1989 and a REMA strip DEM from 2019 near PPI with a background LIMA





**Table 3.** Coverage of DEM over Glaciers, Ice-Free Areas, and Overall for each subset

| Subset | Coverage on Glaciers (%) | Coverage on Ice-free Areas (%) | Coverage, All(Glacier+ Ice-free Areas) (%) |
|---|---|---|---|
| North1 | 36.98 | 40.66 | 37.03 |
| North2 | 27.92 | 69.08 | 28.51 |
| Arrowsmith1 | 27.53 | 56.50 | 29.59 |
| Arrowsmith2 | 28.95 | 54.71 | 30.49 |
| South1 | 29.69 | 55.69 | 35.53 |
| South2 | 30.84 | 66.99 | 34.70 |
| PPI | 42.18 | 48.57 | 43.67 |
| Adelaide Island | 19.51 | 29.75 | 19.88 |

## 4.2 Camera Orientation

### 4.2.1 Exterior Orientation

Exact camera locations and orientations are not available for the IfAG survey. We estimated initial camera positions from the survey index map as mentioned in subsection 3.1. The accuracy of the generated point clouds and the DEMs depends directly on the accuracy of the camera positions. The horizontal camera positions were adjusted on average by up to 2000 m during bundle adjustment and an additional 200 m after applying the transform from the co-registration process, while vertical positions were adjusted on average by up to 100 m and 50 m, respectively (Figure 6). Adjustments varied across regions due to differences in terrain, image quality, and flightline overlap. For the mainland subsets, North1 required the largest horizontal adjustments up to 6500 m due to a tightly grouped initial location estimate retrieved from the survey index map (Figure 6). North2 subset has planimetric camera location corrections up to 2200 m, while South1 and South2 subsets had moderate adjustments of 1200-1800 m. Arrowsmith1 and 2 showed smaller adjustments up to 900–1800 m, benefiting from varying terrain in combination with better image quality. Adelaide Island required horizontal adjustments up to 1800 m, constrained by limited stable areas, while PPI required adjustments of 1200 m. Vertical adjustments followed similar trends, with North1 and North2 requiring up to 80–100 m corrections, while PPI needed only 30–50 m. These adjustments directly influenced the accuracy of the point clouds and resultant DEMs, with larger corrections correlating with higher initial uncertainties in rugged or data-scarce regions.

### 4.2.2 Interior Orientation

We selected the PPI to optimize the unknown interior orientation parameters. The island covers a total of 27 images from two flightlines containing high-quality imagery, facilitating robust bundle adjustment in Metashape. We run image alignment in Metashape with an initial extrinsics estimate (subsection 3.1), a focal length of 85.5 mm, and other processing parameters as mentioned above (Table 1). Allowing the focal length to be optimized during the bundle adjustment resulted in elevation-

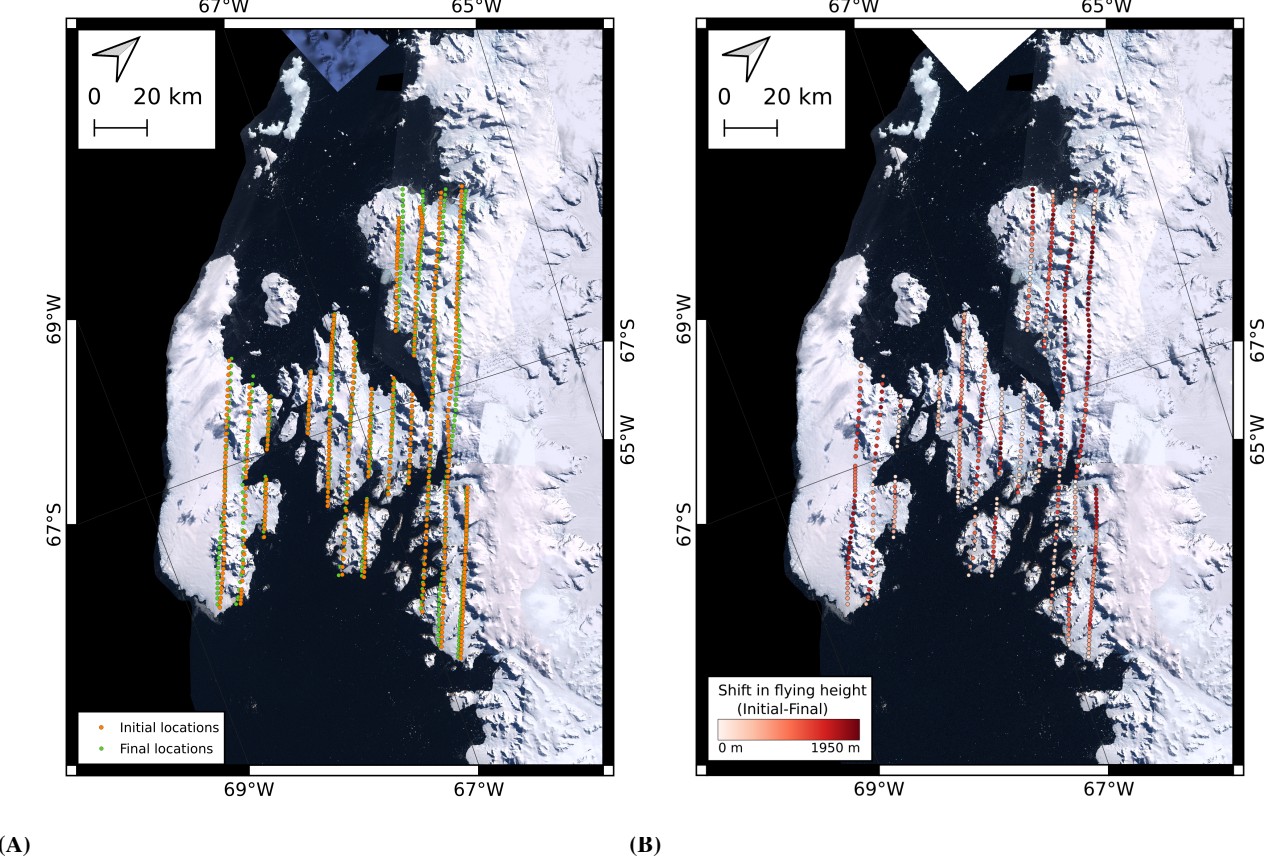

**Figure 6.** (A) Initial and Final(adjusted) camera locations, (B) Difference in Initial and Final(adjusted) camera flying height, with a background LIMA

**Table 4.** Camera calibration parameters

| Parameter | Value |
|---|---|
| Cx, Cy | 6.1890, -4.4350 (pixels) |
| K1 | $1.706 \times 10^{-5}$ |
| K2 | $-1.382 \times 10^{-5}$ |
| K3 | $4.204 \times 10^{-5}$ |
| P1 | $7.624 \times 10^{-6}$ |
| P2 | $4.239 \times 10^{-5}$ |

dependent biases of up to 5 m, likely due to overfitting in areas with sparse tie points. In contrast, using a fixed focal length reduced these biases (as shown in Figure 7). Therefore, we adopted a constant focal length throughout the archive.



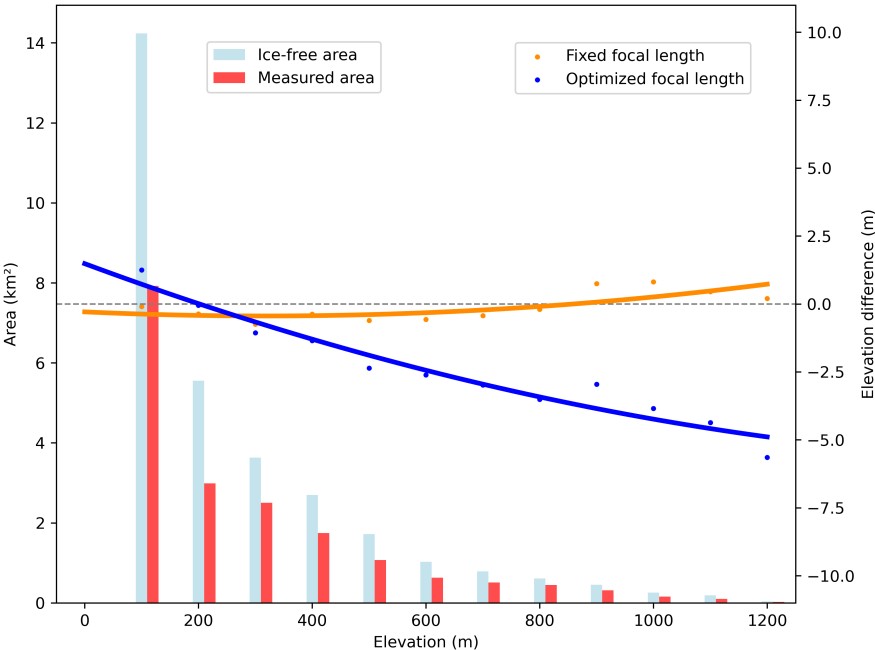

**Figure 7.** Mean elevation dependent bias (right y axis) when focal length is fixed (orange) vs optimized (blue) as a function of altitude bins, lines with respective colors represent a two-degree polynomial fit and ice-free area, measured area, of each bin (left y axis) in light blue, red, respectively.

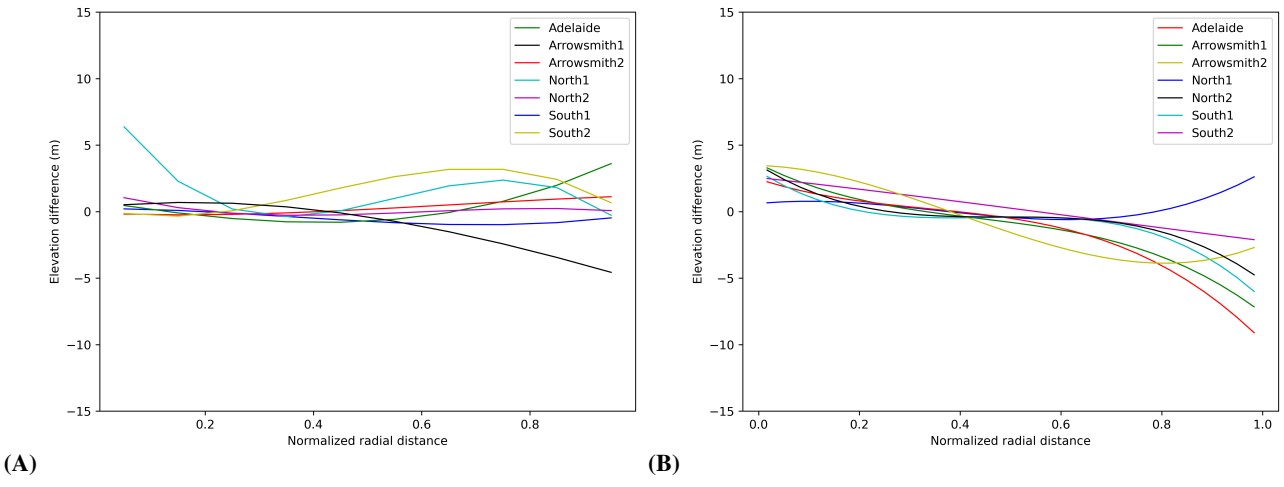

**(A)**                                                                 **(B)**

**Figure 8.** Average elevation bias with respect to (A) Normalized elevation, (B) Normalized radial distance from subset center

The estimated camera intrinsic parameters (Table 4), including radial and tangential distortion, were applied to all subsets and regions. The maximum radial distortion associated with these coefficient values was approximately 8 pixels (0.1 mm at image





**Table 5.** Elevation error statistics for IfAG DEM compared to REMA

| Statistic | 0-10° | 10-20° | 20-30° | >30° | All Slopes |
|---|---|---|---|---|---|
| No. of Observations | 160981 | 294415 | 414546 | 3315259 | 4185201 |
| Mean (m) | -0.64 | -0.39 | -0.28 | 0.51 | 0.32 |
| Standard Deviation (m) | 3.60 | 3.99 | 4.51 | 8.46 | 7.78 |
| Median (m) | -0.25 | -0.22 | -0.08 | 0.60 | 0.35 |
| NMAD (m) | 2.72 | 3.24 | 3.98 | 6.63 | 5.83 |

corners), and the maximum tangential distortion was about 1 pixel (0.0125 mm at image corners), indicating well-constrained lens characteristics. Elevation differences with respect to the reference DEM on stable areas used for co-registration as a function of normalized elevation and radial distance from each of the subset centers were analyzed to assess the camera model's performance. Across all regions, elevation-dependent biases remained within 5 m, with minimal but consistent biases observed in Arrowsmith1, and South2, where stable areas are limited at higher elevations. For the North1 subset, this bias is pronounced

in lower areas too, reflecting the subset-specific challenge of limited availability of stable areas for co-registration in the area. Moreover, biases were most apparent at normalized elevations > 0.6, reflecting challenges in rugged terrain (Figure 8). Average elevation bias with respect to normalized radial distance from subset centers showed errors within 5 m except for Arrowsmith1 and Adelaide Island subsets, where biases slightly exceeded 5 m after a normalized radial distance of 0.8. Most of the subsets, higher elevation errors are observed for the pixels farther from the subset center, constrained by the estimated

distortion parameters.

## 4.3 Evaluation of IfAG DEMs with REMA

### 4.3.1 Pixel-level relative accuracy

We evaluated the accuracy of IfAG DEMs with respect to the reference REMA. DEMs were co-registered to REMA using stable areas with slopes less than 30°. The distribution of elevation differences for our IfAG DEMs on ice-free areas is shown

in Figure 9. The IfAG DEMs have an uncertainty of less than 6m (NMAD of 5.83 m) with negligible biases on 4185201 observations (Table 5). Notably, our DEMs show uncertainty below 5 m for the slopes less than 30°, which is important because only ~10 % of the glacier area in the study region is steeper than this slope threshold (Figure 9). Contrastingly, the uncertainty slightly exceeded 6 m for steeper slopes (NMAD of 6.63 m for slopes > 30°). Photogrammetric processing often fails in steep terrain due to shadows and strongly oblique viewing angles, which result in sparse tie points (Nuth and Kääb,

2011). We further observed error variations in different slope categories, with lower slopes showing lower spread in the error and higher slopes showing higher spread, with 0-10°, 10-20°, 20-30°, >30° slope categories showing NMADs of 2.72 m, 3.27 m, 3.98 m, 6.63 m, respectively (Table 5, Figure 9). To further characterize the DEM uncertainty, in the following section, we examine the spatial correlation error of the DEM uncertainty.





**Figure 9.** (A) Histograms of elevation difference for IfAG DEMs with REMA, (B) Ice-free(red) and Glacier (light blue) areas and Ice-free area elevation difference (blue dots) distributions as a function of slope, error bars represent NMAD of elevation difference values in the individual slope interval. Note: for better representation, Ice-free areas are scaled by a factor of 10.

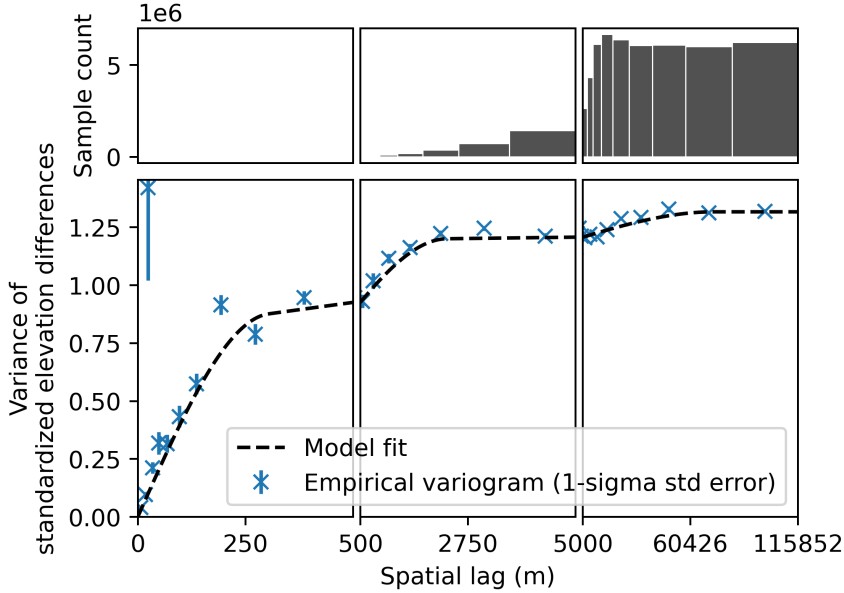

**Figure 10.** Spatial autocorrelation of elevation error of IfAG compared to REMA. Empirical variograms and triple-range variogram model fit of elevation differences on ice-free areas, Short-range: correlation length- 303.97 m, sill- 0.7969 and Medium-range: correlation length- 2312.29 m, sill- 0.3970 and Long-range: correlation length- 72606.89 m, sill- 0.1219

### 4.3.2 Spatially autocorrelated error

We extracted empirical variograms from the elevation difference of IfAG and REMA on ice-free areas to estimate the spatial autocorrelation in our IfAG DEMs (Figure 10). We first standardized the outlier-filtered elevation differences using xDEM's *infer_heteroscedasticity_from_stable* function, which captures spatially heteroscedastic variability by estimating elevation error as a function of terrain slope and maximum curvature (xDEM contributors, 2021; Hugonnet et al., 2022). Using a sample size of 5000 from all our standardized elevation differences in ice-free areas, we sampled and averaged 10 unique empirical vari-

ograms. We used the *infer_spatial_correlation_from_stable* function in xDEM for this purpose (xDEM contributors, 2021). We then fit a triple-range spherical variogram model with short-range, medium-range, and long-range correlation lengths, variance sills of 303.97 m, 0.7969 & 2312.29 m, 0.3970 & 72606.89 m, 0.1219 respectively (Figure 10). The triple-range model effectively captured medium-range correlations in our dataset, likely due to residual camera lens distortion, a double-range model (not shown here, see Figure A2) failed to capture this. The double-range model only accounted for short-range correlations,

attributed to sensor noise, and long-range correlations, stemming from co-registration errors (e.g., misalignment across image subsets) (Dehecq et al., 2020; Hugonnet et al., 2022).





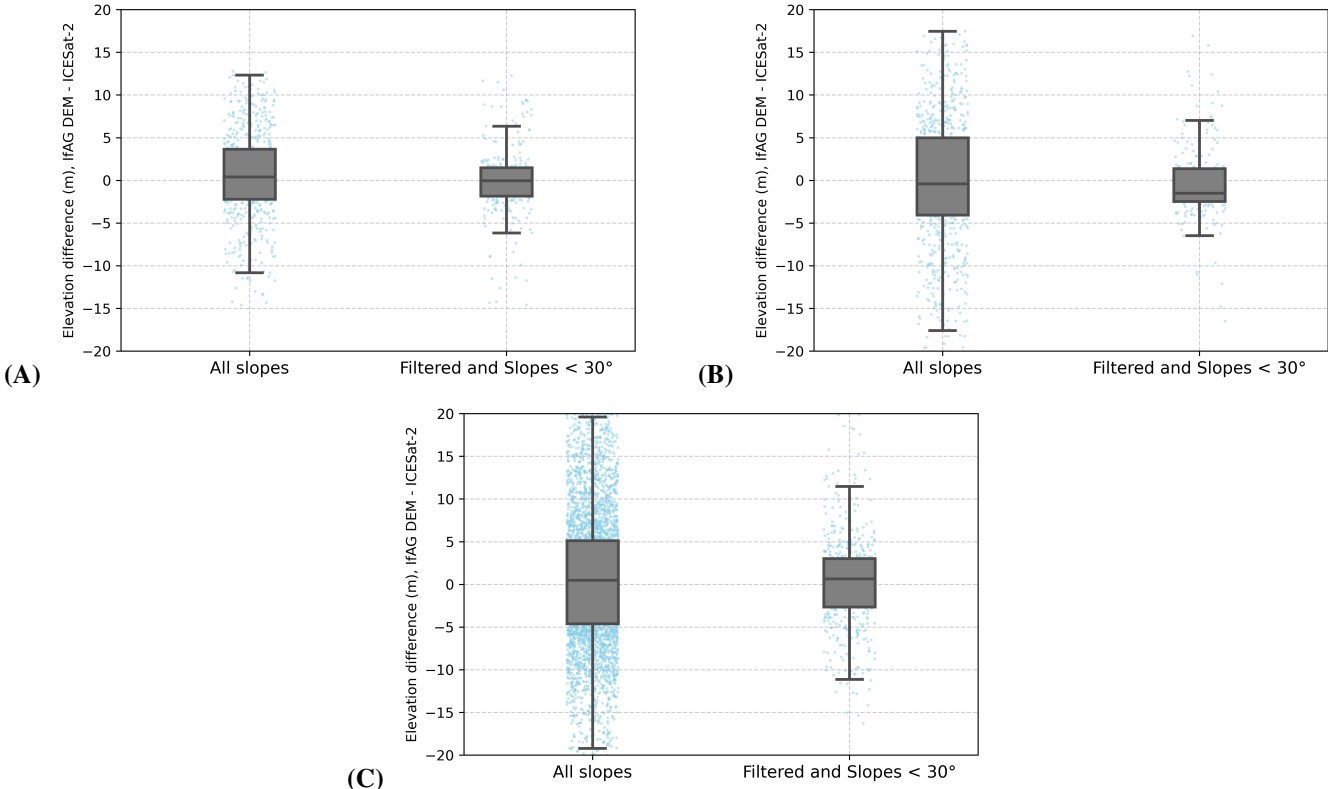

**Figure 11.** Box plot of elevation change with ICESat-2 for (A) PPI, (B) Adelaide Island, and (C) Mainland DEMs.

## 4.4 Evaluation of IfAG DEMs with ICESat-2 data

To evaluate the vertical accuracy of IfAG DEMs with independent surface elevation data, we have taken ICESat-2 ATL06 data from the summers of 2020-21. Around 70000 points are available in the ice-free areas in the study area. We estimated

the uncertainty with respect to filtered ICESat-2 data on 1. Stable areas used for the co-registration , and 2. All ice-free areas. The reduction from around 70,000 to 6,000 (total no.of observations Table 6 ) ICESat-2 points highlights validation challenges (Figure 11). Our conservative outlier filtering (subsection 3.6) removed unreliable ICESat-2 data caused by clouds but also highlights misalignment issues with ICESat-2, likely at steep cliffs (further slope filtering reduced the no.of observations to 1000).

All our DEMs have vertical accuracy with respect to ICESat-2 of less than 8 m (maximum NMAD of 7.21 m for Mainland IfAG DEM). A better accuracy of less than 5 m is observed on stable areas used for co-registration (4.16 m for Mainland IfAG DEM). Due to blunders and outliers on slopes greater than 30 degrees, DEMs have biases up to 0.48 meters (Mean offset for





**Table 6.** Elevation error statistics for IfAG DEMs with respect to ICESat-2 data

| Statistic | Mainland | Adelaide Island | PPI |
|---|---|---|---|
| **All Slopes** | | | |
| No. of Observations | 4,392 | 776 | 619 |
| Mean (m) | 0.48 | -0.10 | 0.66 |
| Standard Deviation (m) | 8.11 | 7.63 | 5.12 |
| Median (m) | 0.49 | -0.40 | 0.41 |
| NMAD (m) | 7.21 | 6.86 | 4.05 |
| **Slopes < 30° and Filtered** | | | |
| No. of Observations | 532 | 217 | 271 |
| Mean (m) | 0.30 | -0.43 | -0.04 |
| Standard Deviation (m) | 5.70 | 4.45 | 4.07 |
| Median (m) | 0.65 | -1.50 | -0.04 |
| NMAD (m) | 4.16 | 2.35 | 2.48 |

IfAG mainland mosaic for all slopes). These biases reduced to 0.3 m (Mean offset for IfAG mainland mosaic for filtered, slopes less than 30 degrees), when only lower slopes are considered (Table 6).

## 4.5 Comparison with other DEMs based on historical aerial imagery

Few DEMs based on similar historical aerial imagery are available for comparison with our IfAG dataset. North and Barrows (2024) recently published an elevation dataset for Larsen B glaciers, derived from 1968 aerial imagery. They reported vertical uncertainties relative to the REMA strip DEMs from 2021 of 15.22 m and 19.21 m for Crane and Flask glaciers, respectively. Another dataset covering the Greenland Ice Sheet, based on 1978–1987 aerial imagery, demonstrated varying vertical
accuracies according to the year of the campaign of up to 8.8 m for slopes < 20° and 10.3 m for all slopes when validated against Airborne Topographic Mapper (ATM) data from 1994–2014 (Korsgaard et al., 2016). Fieber et al. (2018) estimated detailed elevation and volume changes of 16 individual glaciers in northern Antarctic Peninsula using FIDASE archives from 1956-57 austral summers. Using least square surface matching with modern DEMs derived from WorldView-2 imagery, they obtained historical DEMs with post-matching biases varying between +1 m to -5.9 m and uncertainties between 7.3 m to
28.2 m. In contrast, our IfAG DEMs, derived from 1989 imagery and validated against REMA and ICESat-2 data, exhibit lower vertical uncertainties (e.g., Adelaide Island: NMAD 6.39 m with REMA, Mainland: NMAD 7.21 m with ICESat-2 (Figure A1,Table 6)), outperforming the Larsen B datasets by 3-5 times and matching or exceeding the accuracy of the Greenland, northern AP datasets on average.

The larger uncertainties in the Larsen B DEMs may be attributed to the lower quality of the 1968 imagery, poor stereo
overlap, manual tie point and GCP placement, and the lack of precise camera positioning. Similarly, the elevated uncertainties in the FIDASE dataset can be attributed to the age and condition of the 1956–57 film negatives, and differences in type of





scanners used in the digital archiving, which led to scaling issues in these image-derived products (Fieber et al., 2016, 2018). Although both the Larsen B and IfAG datasets lack camera calibration reports and rely on imprecise initial camera positions, our DEMs benefit from a refined estimated camera model and an iterative co-registration approach using the Iterative Closest Point (ICP) algorithm. These methods effectively reduce vertical errors, even for subsets with large initial geolocation offsets (up to 6500 m on North1). Additionally, our DEMs show minimal vertical biases relative to REMA (e.g., PPI: -1.22 m, Mainland: 0.0.49 m, Adelaide Island: 0.82 m,Figure A1), compared to higher biases reported for Larsen B (e.g., Crane: 6.4 m vs. REMA, –4.16 m vs. ASTER; Flask: –0.06 m vs. REMA, –9.01 m vs. ASTER) (North and Barrows, 2024) and FIDASE (up to -5.9 m vs. WorldView-2) (Fieber et al., 2018). While the Greenland dataset benefits from access to camera calibration reports and extensive terrestrial GPS-based ground control, our comparable accuracies were achieved through the co-registration with and validation against spatially well-distributed reference terrain.

## 5 Conclusions and Outlook

We presented a historical DEM and orthomosaics dataset derived from the IfAG aerial imagery archives from 1989 on the western Antarctic Peninsula and surrounding islands. The dataset has been derived using Multiview Structure from Motion (MV-SfM) methods covering 12000 km$^2$ of glacier area. Using initial camera locations from a survey index map and multistage co-registration based on ICP to a reference DEM (REMA), we processed approximately 550 images to produce a historical elevation dataset. Unavailable camera intrinsic parameters are estimated from 27 images from the calibration site, PPI, and used for the entire mission. With coverage on the glacier surfaces varying between 20-42 % , our historical DEMs have vertical accuracies better than 6 m and 8 m when compared to modern elevation data, REMA, and ICESat-2, respectively.

Our dataset is a unique product that aids in the historical glacier monitoring in one of Earth's most rapidly warming, yet data scarce, regions. Combining our dataset with other historical or modern records could provide an unprecedented multi-temporal long-term analysis of glacier volume, area, and mass changes on the Antarctic Peninsula.

*Code and data availability.* The dataset is publicly available at https://doi.org/10.5281/zenodo.16836526 (Thota et al., 2025). The Historical Structure from Motion code is publicly available as a Github package with MIT license at https://doi.org/10.5281/zenodo.5510870 (Knuth et al., 2021a).



# Appendix A

## A1 Evalution of IfAG DEMs with REMA

### A1.1 Pixel-level relative accuracy

**Figure A1.** Histograms of elevation difference for IfAG DEMs on Ice-free areas with A) PPI B) Adelaide Island C) Mainland

## A1.2 Spatial autocorrelation error

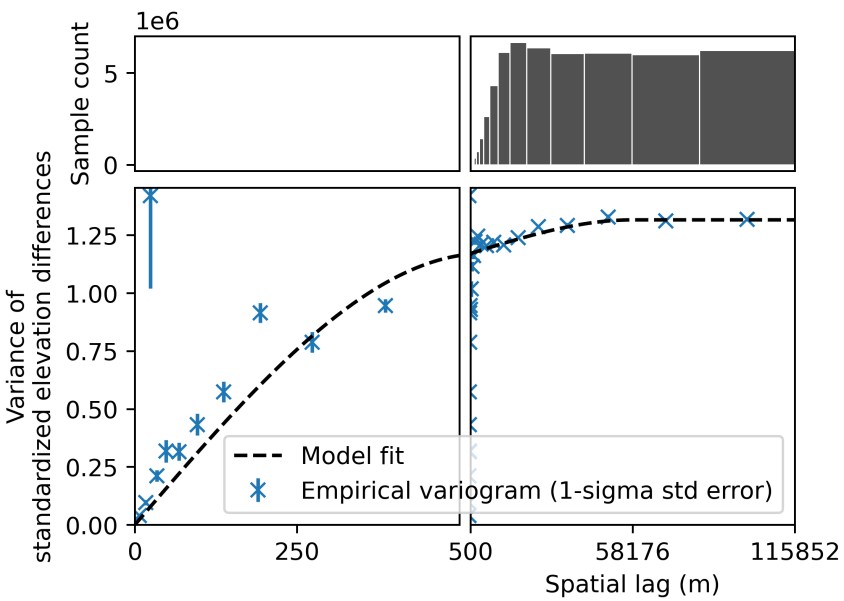

**Figure A2.** Spatial autocorrelation of elevation error of IfAG compared to REMA. Empirical variograms and double-range variogram model fit of elevation differences on ice-free areas, Short-range: correlation length- 538.24 m, sill- 1.1685 and Long-range: correlation length- 58155.99 m, sill- 0.1472

*Author contributions.* VT: Data Curation, Formal Analysis, Investigation, Methodology, Software, Visualization, Writing - Original Draft Preparation, TS: Conceptualization, Data Curation, Funding Acquisition, Project Administration, Supervision, Writing - Review & Editing. FK: Formal Analysis, Software, Writing - Review & Editing. AD: Formal Analysis, Software, Validation, Writing - Review & Editing. CS: Data curation, Writing - Review & Editing. MB: Conceptualization, Funding Acquistion, Writing - Review & Editing

  *Acknowledgements.* The authors would like to thank the AdP for providing the aerial imagery of the AdP collection F 10: Bundesamt
für Kartographie und Geodäsie (BKG). This research has been supported by the European Space Agency (Living Planet Fellowship MIT-AP), the Elitenetzwerk Bayern (grant no. IDP M3OCCA), and the Deutsche Forschungsgemeinschaft (DFG) (in the framework of the priority program SPP1158 "Antarctic Research with comparative investigations in Arctic ice areas" grant no. DFG SE3091/3-1 as well as within the Emmy-Noether-Program grant no. DFG SE3091/5-1). We acknowledge financial support by Deutsche Forschungsgemeinschaft and Friedrich-Alexander-Universität Erlangen-Nürnberg within the funding programme "Open Access Publication Funding". Computing





resources were partly provided by the project EVERLASTING (Erfassung von Vögeln und Meeressäugetieren in Luftbildsequenzen mittels Verfahren der künstlichen Intelligenz) funded by Bundesamt für Naturschutz (BfN) (grant no. 3523820100).

To enhance the language and legibility of the manuscript, the authors used ChatGPT (https://chatgpt.com/) and Grok (https://grok.com/). The output of this service was reviewed and edited by the authors as needed. The authors take full responsibility for the content of the presented manuscript.





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
