# Peer review of "Digital Elevation Models and Orthomosaics of 1989 Aerial Imagery of the Western Antarctic Peninsula and Surrounding Islands between 66-68°S"

_Earth System Science Data, 2025_

## Referee Comment (RC3)

**Review of Thota et al.: "Digital Elevation Models and Orthomosaics of 1989 Aerial Imagery of the Western Antarctic Peninsula and Surrounding Islands between 66-68°S"**

**Summary**

The authors present a novel data set of Digital Elevation Models and Orthomosaics generated from historical aerial imagery collected over the Antarctic Peninsula using a workflow adapted from the Historical Structure from Motion Package (Knuth et al. 2023). The DEMs are calibrated and independently validated against modern high-resolution satellite optical stereo (REMA) and laser altimetry (ICESat-2) observations over non-glacierized areas. The historical dataset will be very valuable in extending the historical mass balance observation record over this data scarce region. I have some comments which the authors can consider to clarify some issues in the manuscript, and I am looking forward to published manuscript post revisions.

**Major Comments**

The manuscript organization can be streamlined with clear division between the methods and results and discussion. There are some places where new experiments are introduced in the results section. I have detailed those at specific occurrences in the general comments below. Reorganizing the sections properly will go a long way in improving the readability of the manuscript.

**General Comments:**

- In the first figure, would it be possible to enlarge the Antarctica-wide subfigure, and add star markers to locations for which previous studies have provided historical mass balance, as described in the introduction literature review? It will provide context to a wider set of audience, and clearly establish the knowledge gap being filled by the new data set and manuscript.
- In auxiliary data section, confirm if the REMA strips are first co-registered and then mosaicking is, or the mosaic tiles are co-registered to CryoSat/ICESat. Currently it is mentioned that the mosaic strips are co-registered to altimetry observations.
- Page 6, Line 140: I am not sure if we can compute base to height ratio/convergence angles from pixel reprojection errors. If the camera parameters are estimated well, the reprojection error can be very well 0. Maybe this was a typo in the sentence, please check and consider modifying.
- First sentence in the paragraph on line 145 is unclear: we should define better what is meant by poorly localized reprojection error, and why were the sizes of these tie points larger than the others which caused this issue. Please clarify and revise.
- The description of the uncertainty estimation between 185 to 190 is a bit repetitive. Consider describing the Seehaus et al. 2019 approach first, which is unchanged for both cases. Then just mention you did a 50 m outlier filter on elevation residual additionally on the altimetry

- measurements before applying the general Seahaus et al. approach to get rid of outliers due to clouds.
- The image quality section in the first result section describing the Shanon index is a bit out of place. It should be mentioned first in the methods. And the results & discussion sections should only describe the results and interpretation, not a new method.
- Why is Section 4.3.1 stated as "relative accuracy"? In general relative accuracy would refer to residual values between maybe the overlapping regions of the historical DEM strips. In comparing against external reference like REMA, should this not be absolute accuracy?
- The description of the variogram analysis could be improved. Like the previous comment, the experiment setup should be mentioned in the method section. In the description section, some interpretation of the the three ranges and sills should be provided. Some speculative points on why was a range upto a particular value on the three different scales, what do we understand from the different sill measurements would be useful.
  Right now there is only a comparison with a double nested spherical model and not much weight is given to the main 3-variogram result presented in the manuscript.
- Reviewer 1 raises a good point about the degraded IS-2 accuracy in areas with high slopes and surface roughness. It would be good to acknowledge this in the discussion section. Also refer to papers by Schenk et al., 2022; Csatho et al. 2024 and other members of IS-2 science team etc on this topic.

Note: I did not download the data for evaluation. This review is for the science manuscript only. As a note, I did not read the already available reviewer comments before I completed the first draft of my review.

Looking forward to the revised manuscript,
Shashank Bhushan

**References**

https://ieeexplore.ieee.org/stamp/stamp.jsp?arnumber=9698042
https://www.sciencedirect.com/science/article/pii/S0034425722004850
https://tc.copernicus.org/articles/13/2537/2019/
https://isprs-archives.copernicus.org/articles/XLVIII-3-2024/83/2024/

---

## Author Comment (AC1)

Dear RC1,

Thank you for reviewing our manuscript and for your constructive and comprehensive feedback. Please find our responses below. To help distinguish between comments and our responses, comments are shown in black and our responses in Blue.

Thota et al., present newly processed DEMs and orthomosaics from 1989 over the western Antarctic Peninsula; a climatically sensitive region in need of high-accuracy datasets to estimate glacier mass change. They use established methods for processing and validating the data, and deliver the dataset in an easily readable format. My assessment of the manuscript is that the work is of high scientific rigour, and my comments are mostly on the presentation of the data. I recommend a round of minor revisions to account for them, and congratulate the authors on a well-designed study.

General comments

I downloaded and visually inspected the DEMs and orthomosaics. The DEM quality seems to vary between being visually excellent to rougher looking in other spots, for example in the north-eastern edge of Adelaide Island (near Hansen Island). I think it would be a great addition to add a layer that could be used to filter these issues out for future uses of the data. For example, a per-pixel point count, standard deviation, or the "confidence" score from Metashape should all probably reveal where bad pixels are, which could be used for filtering by the user. More simply, perhaps just publishing the dense point clouds could be an option too (pre- or post-co-registration).

Thank you for your observation. As you rightly noticed, the quality of the DEMs is generally good, but the effect of stable areas for the coregistration dampened the quality in a few locations, apart from image quality, scale, and overlap. The north-eastern edge of Adelaide Island is related to the issue of insufficient stable areas for co-registrations (can be seen in Figure 3B in the manuscript). We agree that the availability of other data would present the user with a choice to judge or fix the quality issues in the future. Therefore, we decided to add pre-coregistration dense point clouds to the Zenodo data repository.

The use of ICESat-2 ATL06 elevation data as an independent validation method strengthens the case for the accuracy of the newly produced DEMs. I wonder, however, if issues with these data might be even worse than accounted for in the current version of the manuscript, leading to an overestimation of uncertainty in the new data when comparing the two. In other words, I think there is a chance that your data might be better than reported. The MSc thesis by Liu (2023)* details the use of ICESat-2 ATL08 data for snow depth retrieval, and find concerning issues on the accuracy of the product in both high slope and high curvature areas. This is briefly discussed in the published version of his thesis (Liu et al., 2025), but much information was unfortunately lost in the publication process. I am not sure where the slope/curvature errors are introduced; perhaps they are not present in the ATL06 product at all, but I nevertheless

recommend the assessment of not just slope but also planform and profile curvature. For example, binning the elevation difference by planform or profile curvature may reveal strong correlations that may be blamed on ICESat-2, not the newly produced DEMs. I recognize that ICESat-2 validation is not a pivotal part of the manuscript, so I leave the exact handling of my comment in the hands of the authors. I simply want to highlight that there may be a way to argue that your data may be better than reported. A minimum treatment could be to bring up in the text that ICESat-2 struggles in high slope/curvature regions, so the difference spread might get lower with further filtering.

We agree that ICESat-2 may exhibit degraded performance in regions of high slope and curvature, as highlighted in Liu et al. (2025) and Shen, X et al. (2022). As suggested, we computed profile curvature and binned the IfAG – ICESat–2 elevation offsets by curvature magnitude (see Figure R1). The results confirm a clear trend: median offsets increase systematically with positive profile curvature, supporting the reviewer's concern that ICESat-2 errors may contribute to the observed uncertainty estimate of IfAG DEMs in the areas of steep, higher curvature. However, with this information, it is difficult to conclude which of the sources is the major reason for the observed bias.

Here, we would like to highlight that our aggressive filtering helps to eliminate some of these outliers for validation against ICESat-2. As shown in Figure R2, a broader filter (applying a simple elevation difference filter of ±200 m between IfAG DEM - ICESat-2) increased some of the extreme offsets in high-curvature bins (higher median, NMAD), compared to our original multi-step filtering (±50 m, 2–98th percentile, 3×NMAD) (Figure R1).

[Figure]

Figure R1. Elevation difference between IfAG DEM and ICESat-2 on ice-free areas with respect to Profile curvature, with our multistep filtering (described above) applied on elevation differences

[Figure]

Figure R2. Elevation difference between IfAG DEM and ICESat-2 on ice-free areas with respect to Profile curvature, with a simple filtering (described above) applied on elevation differences

Given that ICESat-2 validation is not central to our study, and considering the known limitations of ICESat-2 in high-relief areas, we have chosen to retain our original filtering and reported statistics. However, we added a brief discussion in the revised manuscript (Section 4.4) acknowledging that ICESat-2 performance degrades with increasing curvature, and that the true accuracy of our DEMs may therefore be underestimated in such regions.

We added the following text to L288 of the manuscript,

*"These biases reduced to 0.3 m (Mean offset for IfAG mainland mosaic for filtered, slopes less than 30 degrees), when only lower slopes are considered (Table 6). Furthermore, the accuracy of ICESat-2 is known to degrade at higher curvatures (Shen, X et al. 2022); the uncertainties of our DEMs may therefore be overestimated in such regions."*

Specific comments

L40: What did they find in the "follow-up analysis" of Fieber et al., (2018)? The sentence stops quite abruptly. For example, adding an average geodetic mass balance, such as in the sentence below, would complete it.

We added the findings of the paper to make the sentence meaningful.

*"In a follow-up analysis, Fieber et al. (2018) analyzed the surface elevation changes of 16 individual glaciers, grouped at 4 locations on the AP and surrounding islands, between 1956 and 2014. They reported that 81% of the glaciers exhibited significant thinning, with an average*

*annual mass loss rate of 0.24 ± 0.08 m w.e. Most notably, this was observed at Stadium Glacier, where losses reached up to 62 m w.e. and the glacier front retreated by more than 2.2 km."*

L50: "[…] against external elevation data such as […]"; I recommend being more specific than "such as"; you validate against REMA, ICESat-2 and other published DEMs.

*We updated the sentence to "The DEMs have been co-registered and evaluated against external elevation data such as Reference Elevation Model of Antarctica (REMA), ICESat-2 and other historical DEMs (Howat et al., 2019)."*

Figure 1. Great overview, but the locality labels are very small to the point of being unreadable at 100% A4 zoom. Please also add Adelaide Island and Pourquoi Pas Island, as they are repeatedly mentioned in the text.

We updated Figure 1 according to the review's suggestions.

L80: According to their webpage, it seems that only early REMA releases used ICESat-1 co-registration. As far as I understand, the mosaic is not using these data. However, REMA version 2 seems to be using ICESat-2 and TandemX data for co-registration. I can also not find any mention, apart from the REMA front-page, that discusses Cryosat-2, which is strange on their part. Please adjust the text according to the dataset version that was used. https://www.pgc.umn.edu/guides/stereo-derived-elevation-models/pgc-dem-products-arcticdem-rema-and-earthdem/

Thanks for bringing this out. We used version 2 of REMA in our work, and they seemed to have used ICESat-2 and Tandem-X 90m PolarDEM for the co-registrations as per the cited documentation. We added the following text to reflect this update.

*"We used the Reference Elevation Model of Antarctica (REMA, version 2) mosaic as a reference DEM to extract stable (or static) ground elevation for co-registering our historical DEMs derived from the aerial imagery…….strip DEMs to avoid edge artefacts. REMA mosaic tiles are co-registered to ICESat-2 and Tandem-X 90m PolarDEM."*

L99: Which version of Metashape did you use? I see that this information is provided on L131, but it should be the mentioned the first place where Metashape is mentioned.

Updated.

Sect. 3.1/3.2/Figure 2. I was mildly confused by starting to read about the extrinsic parameter estimation, then reading the workflow of Figure 2 which starts with intrinsic parameter estimation (specifically fiducial estimation). I think it would be easier to read if the camera intrinsic estimation section (3.2) came before the extrinsic (3.1) to stay consistent with the figure.

Thanks for your suggestion. We have updated the sequence of sections 3.1, 3.2.

Figure 2: Generally a great figure! It took me some passes, however, to understand that the lines without arrow signs were detailed explanations of whatever they were connected to. I especially got lost in the multi-stage co-registration as my eyes were flying back and forth between all the arrows when and I tried to logically arrive to the ICP box. I have no great suggestion for how to fix the clarity of the detail boxes (perhaps like chat boxes in comics?), but I suggest a small revision to the styling. I see that the boxes are rhombohedral to separate "parameter" boxes, but two of them are not ("OpenCV matchTemplate" and "Key point identification […]"). Another alternative could be to color their background differently. Finally, "Nuth and Kaab" should be "Nuth and Kääb".

Thanks for your perspective. We updated the figure in the revised manuscript accordingly.

L110: What happened to the images with one or zero principal points? Were they discarded? I suggest adding a short sentence on that here.

We assume that you meant one or zero 'fiducial' points, as clarified in L113-114; we discarded all the images with fewer than two fiducial points detected from further analysis.

L111: The text states that 29.50% of all images had three fiducial markers detected, then that the principal point is extracted from the centroid of the detected fiducials. I hope you mean the centroid of two opposing fiducials in the case of three detections! Otherwise, the estimated principal point would be very far off from the real one. I suggest phrasing it to make sure that you've thought of the case of three fiducials.

Thank you for this careful observation. We take this opportunity to clarify. When three fiducials are detected, the principal point is computed as the midpoint of the two opposing fiducials that form a complete pair (e.g., left–right). The third detected marker is not used in this case.

We updated the text to clarify this.

*"The principal point was estimated from fiducial markers that passed quality checks. It was computed as the average of axis-aligned pairwise midpoints (left–right and/or top–bottom). For instances where only two non-opposing markers were detected (e.g. left and bottom), the principal point was estimated using the X coordinate from either the top or bottom marker and the Y coordinate from either the left or right marker (HIPP, 2021)."*

L120: Missing "a" in "[…] Pourquoi Pas Island (PPI, see Figure 3) as calibration site."; "a calibration site".

Corrected.

L121: Why use only one site for intrinsic calibration? I don't understand why they were not estimated over the entire survey instead. I'm sure there's a good reason, but I don't learn it by reading the paragraph. Please add a sentence why you did this.

Thanks for this great question. Intrinsic calibration is performed at only one site because only one camera is used for the entire survey. Moreover, we chose this particular site for optimising the intrinsic parameters of the camera, mainly for two reasons: 1. It has well-spread stable areas for optimising intrinsics, given that initial camera locations are not accurate, and 2. It has high image quality compared to other regions of the study area. We have updated L120-124 to the following

*"To estimate the intrinsic parameters of the single camera used throughout the survey, we performed camera calibration at Pourquoi Pas Island (PPI, see Figure 3). This site was selected for two main reasons: (1) it contains well-distributed, stable terrain with significant terrain features representative of the broader Antarctic Peninsula, enabling robust parameter estimation despite initial positional inaccuracies (Cziferszky et al., 2010), and (2) it offers the highest image quality in the dataset, with cloud-free coverage and strong visible contrast."*

Table 2: Very interesting relationships between coverage, resolution and uncertainty. Thank you for informing about that! Small technical correction: the quality flag is "Ultra High", not "Ultrahigh".

Thank you. We corrected the text.

L145: Pedantic comment from me: Technically, are the tie points not used to inform the intrinsic/extrinsic estimation, which in turn allows for a generation of a dense point cloud? Currently, it sounds like the tie points are used to generate the dense cloud, but that is not exactly true as far as I understand it.

Thank you for this helpful clarification. You are correct, the tie points are used to estimate the intrinsic and extrinsic camera parameters, which then enable the generation of the dense point cloud through the dense reconstruction algorithm of Metashape. We revised the text to,

*"The filtered tie points were used to estimate the intrinsic and extrinsic camera parameters, which were then applied to generate a dense point cloud for each subset at medium quality, i.e., at a scale sixteen times lower than the original image scale."*

L146: Here it says that "medium" quality represents 1/16 image scale, while Table 2 states that medium is equivalent to a scale of 1/4.

It is a mistake in the table; 'medium' quality represents 1/4 on each side of the image, therefore, 1/16 image scale. We corrected this information in Table 2

L150: I suggest adding "originally" or something to the resolution parenthesis "(~3.5 m)". It took me a while to understand how that aligned to the 10 m REMA resolution, before realizing that I had misunderstood the sentence.

We updated the text accordingly.

L162: "refined the alignment" → "the alignment is refined"

Updated.

L168: I would add one or half a sentence about why sub-pixel co-registration is required after ICP. I know that it's required because minimum distances will always converge to an exact pixel offset when co-registering regular grids, but the reader might not know that (no need to be elaborate, but I'm just pointing out that it's not obvious). Also, which tool did you use for the Nuth and Kääb (2011) implementation? xDEM? If so, then please state that.

Thanks for raising this point. We have used the demcoreg tool for Nuth and Kääb (2011) implementation. We updated the text to,

*"We used the Nuth and Kääb (2011) algorithm from the demcoreg package for subpixel coregistration over stable areas, which has a higher accuracy compared to ICP, as demonstrated by a reduction of NMAD after coregistration (Shean et al., 2021). This method estimates and corrects systematic offsets by relating elevation differences to terrain slope and aspect."*

L197: As far as I can tell, this is the first time "other historical DEMs" are mentioned. Please make sure to mention this earlier in the text (c.f. my comment of L50).

Updated.

Figure 4. Please consider looking over the caption of this figure. The second sentence starts with "One subset on […]" and it is unclear what this refers to. "with a background LIMA" (and no period) could be rephrased to be more clear. Also, the latitude/longitude labels on the right hand side of the figure overlap so they cannot be read.

Thank you, it is a mistake. We corrected it.

Figure 5. Please define "with a background" better . There is also a missing period after the description of panel B. See my comment on Figure 4 of "with a background LIMA". I think it would be nice to add a brief comment if the REMA strip is the cause for gaps or the IfAG DEM. I presume the latter?

We updated the figure caption. You are correct, the gaps are from the IfAG DEM. We have now added it to the text.

Sect. 4.2 / 4.2.1 / 4.2.2 headers: I suggest changing the header name to something with the word "uncertainty" or "accuracy" in it to more properly reflect its contents.

We have updated the headers accordingly.

L229: It is unclear what you mean with the adjustments directly influencing the accuracy of the point clouds means. Please rephrase this sentence. Do you mean that it significantly improves or harms the accuracy?

Thank you for pointing this out. We have rephrased the sentence to clarify that the adjustments improve the accuracy of the point clouds and DEMs. Larger adjustments indicate areas with higher initial uncertainties, and applying these corrections reduces errors in those regions. We updated the text to,

*"Applying these corrections improved the accuracy of the point clouds and resulting DEMs, with larger adjustments corresponding to regions that initially had higher uncertainties due to rugged terrain or limited data coverage."*

Table 4: Are you sure that the focal length unit is in pixels and not millimeters? Metashape usually translates to millimeters if you use their fiducial-aware features.

We did not use the "film camera with fiducial marks" option in Metashape because we were working with preprocessed images from HIPP (HIPP, 2021). Therefore, the Metashape takes focal length in pixels.

To clarify this, we updated L138 in the manuscript to

*"Preprocessed images were imported into Metashape, and the tie points were generated at the native resolution of the imagery, facilitating precise and independent alignment of each subset."*

Figures 7 and 8: Please add that these are comparisons to REMA (I presume) in the captions.

Updated.

Table 5: I find it strange that no bias is zero since co-registration has been perfomed. It also does not look like that all <30° slope differences would even out to 0 since all medians and means are negative. Is the mask different from the co-registration mask, or does the Nuth and Kääb implementation not include bias-correction (the xDEM implementation does, which is why I ask)? Please help me and future readers to understand the discrepancy!

*We appreciate this insightful comment. As rightly noted, the co-registration surfaces used are different from the mask used to report the uncertainties in Table 5. There are false positives in the existing ice-free area/rock outcrop masks that we filtered to perform the co-registration of our historical DEMs (as explained in Section 3.4 of the manuscript). But, we chose to report the uncertainty of the product over unfiltered, raw ice-free areas from Silva et al. (2020), to stay consistent with previous studies in this region.*

*We would like to clarify that our Nuth and Kääb (2011) implementation through demcoreg package does include bias correction. Nevertheless, the biases may not be zero after co-registration in places where there is limited availability and spatial distribution of stable areas (and may also be due to relatively lower quality historical DEMs). Therefore, the reported uncertainties should be interpreted in light of the spatially variable reliability of the underlying ice-free area masks.*

Figure 9: I would rephrase the end note to be even clearer that it's just scaled for visualisation. For example, it could be "For clearer visualisation, ice free […]" or something similar. Another alternative could perhaps be to change the area unit to normalized area per category, as it may be even clearer then; I leave this up to the authors to decide.

*We updated it as follows "For better visualisation, ice free…"*

Figure 10 caption: The numbers for the variogram model fit are presented both in Sect. 4.3.2 and in the caption. Since they are quite difficult to read out, I wonder if one of them could be removed for clarity. If you want to keep them, then I suggest clarifying the reporting: e.g. what is the hyphen for in "length- 303.97 m"? Perhaps change it to "of": "length of 303.97 m".

*Thanks for pointing it out, updated.*

L274: Replace comma with a semicolon: "[…] camera lens distortion; a double-range model […]".

*Updated.*

L281: Some words are missing in the parenthesis. Perhaps change to "(see the total number of observations in Table 6)"

*Updated.*

**References**

Liu, Z., Filhol, S. and Treichler, D., 2025. Retrieving snow depth distribution by downscaling ERA5 Reanalysis with ICESat-2 laser altimetry. Cold Regions Science and Technology, p.104580.

Shen, X., Ke, C.Q., Fan, Y. and Drolma, L., 2022. A new digital elevation model (DEM) dataset of the entire Antarctic continent derived from ICESat-2. Earth System Science Data, 14(7), pp.3075-3089.

---

## Author Comment (AC2)

Dear RC2,

Thank you for reviewing our manuscript and for your constructive and comprehensive feedback. Please find our responses below. To help distinguish between comments and our responses, comments are shown in black and our responses in Blue.

The manuscript presents a valuable and well-executed study that reconstructs high-quality Digital Elevation Models (DEMs) and orthomosaics from 1989 aerial imagery over the western Antarctic Peninsula. I find this to be a very strong and useful paper; not only because of the released dataset, which fills an important temporal gap, but also because the workflow serves as an excellent methodological guideline for others processing historical aerial data. I recommend minor revisions to improve clarity and consistency in a few sections, but overall the work is scientifically sound and ready for acceptance after minor adjustments.

General Comments

- The calculation of Shannon entropy per image is an interesting quality metric. However, it is unclear whether this value was further used for image selection, filtering, or weighting, or if it is purely descriptive. Please clarify its role in the processing chain.

Thank you for this important point. We used Shannon entropy as a descriptive measure to see if we could explain the coverage variations in our DEMs. However, we also excluded the images with low entropy from the western part of Adelaide Island from our analysis. We have updated L199 of the manuscript to make this aspect clear as follows,

*"We estimated Shannon entropy for each image as an indicator of texture. This was used to assess whether variations in DEM coverage were related to image texture and to filter out low-texture images prior to SfM processing."*

Furthermore, to improve clarity, we have now removed the following sentence from **Section 3.3 DEM generation**

"We excluded images from the western part of Adelaide Island, due to the lack of stable areas and insufficient image features, and from north of Adelaide Island near Grandidier Channel, where images predominantly cover water pixels (Figure 1)."

and added an updated sentence in **Section 4.1 Image Quality and Coverage**

*"Overall coverage (Glacier + Ice-free Areas)........on-glacier coverages spanning 20-42 %. We excluded images from the western part of Adelaide Island due to the lack of stable areas and insufficient image features (low entropy; Figure 4), and from north of Adelaide Island near the Grandidier Channel, where images predominantly cover water pixels (Figure 1). Notably, DEMs for PPI and North1, South1……."*

- The LIMA mosaic is visually appealing but reduces the readability of the point symbols and subset boundaries. Consider replacing it, for example, with the high-resolution coastline polygons. The subset colours are also difficult to distinguish; labelling them (e.g., A–H) or adding clearer outlines would improve readability.

Thank you for your opinion about the visual appearance of the background used in the figure. We updated the figure with your suggestion.

- Will the raw digitized images or their metadata (e.g., image positions, flightlines, scale information) be publicly available?

We decided to publish the pre-coregistration point cloud data, flightlines with image positions in the Zenodo repository. However, the raw images can be obtained from the Archive for German Polar Research (Archive für deutsche Polarforschung - AdP) at the Alfred Wegener Institute (AWI) in Bremerhaven, Germany. We have now included the following information in the Data Availability section,

*"The aerial images used in this study from the 1989 IfAG survey are open to everyone and can be obtained from the Archive for German Polar Research (Archive für deutsche Polarforschung - AdP) at the Alfred Wegener Institute (AWI) in Bremerhaven, Germany."*

Specific Comments:

P1 L3: Spelling Error: acquired by the (...), which is kept in the

Corrected.

P3 F1: Spelling Error: Fiducial mark is spelled wrong (the i is capitalized)

Corrected.

P3 L54: Please specify whether the imagery is entirely nadir or includes oblique frames? * I see it's mentioned in the previous chapter, but please restate it here again for better understanding.

Yes, the imagery was intended to be entirely nadir-looking. We updated the text in the manuscript as follows,

*"The archive consists of approximately 2000 vertical aerial images, acquired during a photogrammetric survey by the Institut für Angewandte Geodäsie (IfAG), Frankfurt am Main, Germany."*

P3 L58: Is it possible to replace "most of the images" with precise percentages?

We updated the text to include the percentage of the images,

*"Approximately 61% of the images were acquired at an average flight elevation of 5895 m, yielding a nominal photoscale of 1:70,000 with forward overlap of about 60%."*

P4 L84: Provide the version and citation of the ADD rock mask used.

We updated the text to *"Glacier outlines, ice-free areas, and rock outcrops are taken from the Silva et al. (2020) and Antarctic Digital Database (High resolution vector polygons of Antarctic rock outcrop V7.3 (Gerrish et al., 2020))"*

P4 L99: Wouldn't it be possible to extract at least an approximate yaw based on the position of the images and the following flightpath?

Thank you for the suggestion. While it is theoretically possible to estimate the yaw based on the flight path and rough estimates of the position of the images, we intentionally provide an initial yaw of 0° with a loose accuracy of 180° to Metashape. This allows the software's Structure-from-Motion (SfM) bundle adjustment to freely estimate the true yaw (along with pitch and roll) during photogrammetric processing. Metashape robustly refines camera orientations using image overlap and feature matching, making pre-computed yaw estimates unnecessary at this stage.

P5 F2: Distinguish the process boxes from technical detail boxes (e.g., OpenCV matchTemplate) using for example italics.

We updated the figure accordingly.

P5 L107: Please rephrase to make it clearer that the median fiducial position is computed per flightline, not across all images.

We updated the text to "....the median position of all matches for each fiducial marker, computed per flightline."

P6 L109: Spelling error: 29.50% of images had three fiducial markers

Corrected.

P6 L112: Please make clearer that at least two fiducial marks from different axes are required to compute the principal point.

Thank you for this helpful comment. You are correct, at least two fiducial marks from different axes (left-right and/or top-bottom) are required to compute a valid principal point. We have revised the text for clarity as follows,

*"The principal point was estimated from fiducial markers that passed quality checks. It was computed as the average of axis-aligned pairwise midpoints (left–right and/or top–bottom). For instances where only two non-opposing markers were detected (e.g. left and bottom), the principal point was estimated using the X coordinate from either the top or bottom marker and the Y coordinate from either the left or right marker (HIPP, 2021)."*

P6 L112: Clarify whether the principal point is determined by intersection or by averaging?

As explained above, we used the "average". We updated the text to clarify this.

P6 L131: Since the total number of processed images should be known precisely, please provide the exact figure instead of an approximation.

We updated the text.

P6 L131: Please clarify whether the subsets were processed as independent Metashape projects, as separate chunks within a single project, or as one project later divided into eight exported subsets.

We processed each subset independently. We updated the text.

*"We processed approximately 550 images from 12 flightlines photogrammetrically in Agisoft Metashape version 2.1.1 in 8 different projects (Figure 4)."*

P7 T1: The table also includes parameters related to alignment and export. Consider expanding the caption to reflect that it lists both photogrammetric and processing parameters. In addition, please specify whether any tie points were masked or filtered (e.g., stationary tie points or water areas).

We updated the caption to *"Photogrammetric and processing parameters used in Agisoft Metashape for image alignment and dense point cloud generation."*
We have used the option of "Exclude stationary tie points", we updated this in the table to reflect this.

P7 T2: Very interesting table. How are the DEM resolutions derived? Aren't these chosen by the user?

Thanks for your comment. You are correct, the resolution of the DEMs can be manually set. In our case, the resolutions of the DEM are determined by 3-5 times the effective ground sampling distance (different at different quality levels) of the input images (Shean et al. 2016). This

reduces the interpolation artefacts in the analyzed DEMs, which helps us to compare different depth quality settings with respect to coverage. We have already clarified this aspect in L150 of the manuscript,

*"...resolution corresponds to approximately three times the effective GSD of the input images processed…"*

P8 L162: In my experience, the rock outcrops from the ADD rock outcrop mask can contain inaccuracies. Did you check for their quality in that region?

That is true. We also experienced the same, we found inaccuracies in the ADD rock outcrop in the studied region. Therefore, we used ice-free areas from Silva et al. 2020 as a major source for the coregistration surfaces (please check our response below on your comment P9 F3).

P8 L167: As this is the first reference to the Nuth and Kääb (2011) method, consider adding one sentence summarizing its function

Thanks for this suggestion. We updated the text to *"We used the Nuth and Kääb (2011) algorithm from the demcoreg package for subpixel coregistration over stable areas, which has a higher accuracy compared to ICP, as demonstrated by a reduction of NMAD after coregistration (Shean et al., 2021). This method estimates and corrects systematic offsets by relating elevation differences to terrain slope and aspect."*

P8 L177: Spelling Error: and the 3D shift

Corrected.

P8 L187: The sentence in its current form is difficult to follow; please rephrase for clarity

We updated this sentence as *"The uncertainty of the IfAG DEMs is assessed using ICESat-2 data by analyzing the distribution of elevation differences between the two datasets over stable areas."*

P8 L182: Please place "REMA mosaic" in parentheses to make clear that this refers to the reference DEM.

We will update the text to "....(REMA mosaic)..."

P9 F3: Only ice-free areas are displayed. Is there a reason the rock-mask extent is not also shown? Including both could improve transparency of the stable-area selection.

We appreciate the suggestion. As noted earlier, we compared the ice-free areas from Silva et al. (2020) with the rock-outcrop mask from the ADD and found that the latter contains some false positives. For this reason, we used the Silva et al. (2020) ice-free areas as a major source for

the coregistration surfaces. The ADD rock-outcrop mask was only used in a few manually selected locations (e.g., in the northern part of the Arrowsmith 2 subset). Therefore, to highlight this aspect, Figure 3A shows only the ice-free areas from Silva et al. (2020), and Figure 3B shows the actual coregistration surfaces used in our study.

P11 F4: Right-side coordinates overlap; adjust spacing.

Thanks for noting this. We updated the figure.

P13 T3: It may be interesting to also include the percentage of glacier and ice-free terrain within each subset.

We appreciate the suggestion. We updated the table as per your suggestion.

P16 L246: The text refers to normalized elevation values, but these are not visible in Figure 8. The caption of panel A seems inconsistent. Please verify and correct.

Thanks for noting this. We updated Figure 8A.

P19 L281: The sentence is not complete.

We updated the text to *"The ICESat-2 validation dataset was reduced from ~70,000 to ~6,000 points after outlier filtering, highlighting the validation challenges in a complex terrain (Table 6, Figure 11)."*

P19 L281: Spelling Error: nr. of observations (space is missing)

Corrected.

P19 L283: ibid.

We presume this comment is incomplete. However, we rechecked L283.

P20 L288: Spelling Error: These biases reduce to

Corrected.

**References**

Gerrish, L., Fretwell, P., & Cooper, P. (2020). High resolution vector polygons of Antarctic rock outcrop (7.3) [Data set]. UK Polar Data Centre, Natural Environment Research Council, UK Research & Innovation. https://doi.org/10.5285/cbacce42-2fdc-4f06-bdc2-73b6c66aa641'

Shean, D.E., Alexandrov, O., Moratto, Z.M., Smith, B.E., Joughin, I.R., Porter, C. and Morin, P., 2016. An automated, open-source pipeline for mass production of digital elevation models (DEMs) from very-high-resolution commercial stereo satellite imagery. ISPRS Journal of Photogrammetry and Remote Sensing, 116, pp.101-117.

---

## Author Comment (AC3)

Dear RC3,

Thank you for reviewing our manuscript and for your constructive and comprehensive feedback. Please find our responses below. To help distinguish between comments and our responses, comments are shown in black and our responses in Blue.

**Summary**

The authors present a novel data set of Digital Elevation Models and Orthomosaics generated from historical aerial imagery collected over the Antarctic Peninsula using a workflow adapted from the Historical Structure from Motion Package (Knuth et al. 2023). The DEMs are calibrated and independently validated against modern high-resolution satellite optical stereo (REMA) and laser altimetry (ICESat-2) observations over non-glacierized areas. The historical dataset will be very valuable in extending the historical mass balance observation record over this data scarce region. I have some comments which the authors can consider to clarify some issues in the manuscript, and I am looking forward to published manuscript post revisions.

**Major Comments**

The manuscript organization can be streamlined with clear division between the methods and results and discussion. There are some places where new experiments are introduced in the results section. I have detailed those at specific occurrences in the general comments below. Reorganizing the sections properly will go a long way in improving the readability of the manuscript.

Thank you for this comment about the readability of the manuscript. We originally chose to describe some of the methods in results and discussion to maintain a concise presentation, highlighting only DEM generation in the methods section. However, we agree that the manuscript may benefit from a clearer separation between the description of the methods and the presentation and interpretation of the results. We have therefore reorganized the manuscript by separating the Methods section while retaining a combined Results and Discussion section to improve readability and avoid unnecessary repetition.

**General Comments:**

• In the first figure, would it be possible to enlarge the Antarctica-wide subfigure, and add star markers to locations for which previous studies have provided historical mass balance, as

described in the introduction literature review? It will provide context to a wider set of audience, and clearly establish the knowledge gap being filled by the new data set and manuscript.

Thanks for this great suggestion. We updated the figure accordingly.

• In auxiliary data section, confirm if the REMA strips are first co-registered and then mosaicking is, or the mosaic tiles are co-registered to CryoSat/ICESat. Currently it is mentioned that the mosaic strips are co-registered to altimetry observations.

Thanks for this comment. We used the REMA v2 mosaic downloaded from the OpenTopography website (Howat et al. 2022). According to the documentation provided by the PGC, REMA mosaic v2 is created to form 50 km x 50 km tiles from repeat strip DEMs with a median value of available elevation values at each pixel, with filtering applied to remove outliers. The tiles are then aligned to ICESat-2 and Tandem-X 90m PolarDEM data (not Cryospat/ICESat). Additional details on the mosaicking procedure are described in Howat et al. (2019).

We have updated L75-81 to clarify this as follows.

*"We used the Reference Elevation Model of Antarctica (REMA, version 2) mosaic as a reference DEM to extract stable (or static) ground ……… derived from the aerial imagery (Howat et al. 2022). The REMA mosaic was downloaded from the OpenTopography portal (https://portal.opentopography.org/). It is compiled from multiple REMA strips that are generated using very high resolution (0.32 to 0.5 m) WorldView-1,2,3 and GeoEye-1 satellite imagery through Surface Extraction from TIN-based Searchspace Minimization (SETSM) software (Howat et al., 2019). The mosaic is created to provide a more consistent and complete DEM product with blending and feathering of strip DEMs to avoid edge artefacts. REMA mosaic tiles are co-registered to ICESat-2 and Tandem-X 90m PolarDEM."*

• Page 6, Line 140: I am not sure if we can compute base to height ratio/convergence angles from pixel reprojection errors. If the camera parameters are estimated well, the reprojection error can be very well 0. Maybe this was a typo in the sentence, please check and consider modifying.

Thank you for spotting this, it was a mistake. The threshold of 10 refers to Metashape's "Reconstruction Uncertainty", not reprojection error (which is indeed unrelated to base-to-height ratio and can be driven to near zero with good camera calibration and optimization). In Metashape, reconstruction uncertainty is defined based on the geometric relationship of the cameras used to generate tie points, and high reconstruction uncertainty values indicate points reconstructed from images with a small baseline (i.e., small base-to-height ratio or small parallax angle) (Metashape manual, version 2.2, see references). We have rewritten the sentence to correct "reprojection accuracy" with "reconstruction uncertainty".

• First sentence in the paragraph on line 145 is unclear: we should define better what is meant by poorly localized reprojection error, and why were the sizes of these tie points larger than the others which caused this issue. Please clarify and revise.

Thank you for highlighting this. We agree. "Poorly localized projections" are the tie points whose image positions cannot be determined by Metashape precisely. This may happen when the feature used as a tie point appears large, blurry and less distinct than other tie points. This makes the average scale that was used for measuring coordinates of the projections of the tie-point in all overlapping images larger, degrading the "Projection Accuracy" measure in Metashape. Therefore, we removed all the tie points with a projection accuracy value of more than 5.

We revised the text to make this aspect clear,

*"We filtered out tie points with low projection accuracy caused by their poor localization. Tie points are poorly localized when the features they represent are large or less distinct, making it harder to locate their exact position in the images. To remove these points, we applied a projection accuracy threshold of 5, which was measured as the average image scale of the feature across overlapping images."*

• The description of the uncertainty estimation between 185 to 190 is a bit repetitive. Consider describing the Seehaus et al. 2019 approach first, which is unchanged for both cases. Then just mention you did a 50 m outlier filter on elevation residual additionally on the altimetry measurements before applying the general Seahaus et al. approach to get rid of outliers due to clouds.

We would like to clarify that our uncertainty estimation with respect to ICESat-2 differs from the procedure used for REMA (and therefore the approach of Seehaus et al. 2019). In addition to applying a 50 m outlier filter to remove clouds, we also applied 2-98 percentile data and a 3 times NMAD threshold on the whole dataset without slope binning. This results in a conservative removal of outliers tailored to the sparse distribution of ICESat-2 elevation measurements.

• The image quality section in the first result section describing the Shanon index is a bit out of place. It should be mentioned first in the methods. And the results & discussion sections should only describe the results and interpretation, not a new method.

We agree. Now, we chose to present the description of Shannon entropy definitions in methods section and the results (along with their discussion) in separate sections to improve clarity and structure.

• Why is Section 4.3.1 stated as "relative accuracy"? In general relative accuracy would refer to residual values between maybe the overlapping regions of the historical DEM strips. In comparing against external reference like REMA, should this not be absolute accuracy?

Thank you for this important point. We chose the term "relative" mainly because the reference DEM itself carries spatially variable uncertainties. The vertical accuracy of REMA is varying and depends on terrain characteristics, image geometry, and processing conditions (Howat et al., 2019). Since REMA is not an error-free "truth" surface, differences against it are best interpreted as accuracy relative to the reference DEM rather than true absolute accuracy.

In addition, the subsection title "**4.3 Evaluation of IfAG DEMs with REMA**" explicitly indicates that this assessment is in reference to REMA. We therefore believe it should be clear that the term relative accuracy refers to accuracy relative to the REMA, rather than to internal relative accuracy derived from overlapping historical strips.

To avoid any ambiguity and to explicitly caution readers, we have added the following sentence at L186,

*"Note that these uncertainty estimates relative to REMA include potential errors present in REMA itself (Howat et al., 2019). Therefore, to obtain an independent estimate of vertical accuracy, IfAG DEMs were also compared to ICESat-2. The uncertainty of the IfAG DEMs is assessed using ICESat-2 data by….."*

• The description of the variogram analysis could be improved. Like the previous comment, the experiment setup should be mentioned in the method section. In the description section, some interpretation of the the three ranges and sills should be provided. Some speculative points on why was a range upto a particular value on the three different scales, what do we understand from the different sill measurements would be useful. Right now there is only a comparison with a double nested spherical model and not much weight is given to the main 3-variogram result presented in the manuscript.

Thank you for this great comment. We agree that the description of the variogram analysis can be clarified and better integrated into the manuscript. In the revised version, we have moved the description of the experiment setup to the Methods section. In the Results and Discussion section, we now provide a more detailed interpretation of the three variogram ranges and sills, including what they imply about spatial correlation at different scales. Following is the updated text about its description in **Results and Discussion**

*"We fitted a triple-range spherical variogram model to characterize the spatial autocorrelation of elevation error in our IfAG DEMs (Figure 10). Each nested component in the variogram fit represents a distinct contribution, with its sill indicating the proportion of total error variance associated with that spatial scale. The short-range correlation (range of 303.39 m, sill of 0.7969) accounts for the largest share of variance, suggesting that most elevation error arises from local sources such as sensor noise. The medium-range component (range of 2312.29 m, sill of 0.3970) contributes the next major fraction of the variance and likely reflects residual lens distortion that introduces correlated errors over several kilometres (Dehecq et al., 2020). A double-range model failed to capture this substantial medium-scale variance (see Figure A2), so*

*we adopted a triple-range spherical model to represent this physically interpretable structure, consistent with known error sources. The smallest proportion of error variance is associated with the long-range correlation (range of 72606.89 m, sill of 0.1219), which reflects broad regional variances caused by co-registration errors, such as misalignment across image subsets (Dehecq et al., 2020; Hugonnet et al., 2022)."*

• Reviewer 1 raises a good point about the degraded IS-2 accuracy in areas with high slopes and surface roughness. It would be good to acknowledge this in the discussion section. Also refer to papers by Schenk et al., 2022; Csatho et al. 2024 and other members of IS-2 science team etc on this topic.

Thank you. We have now updated the L228,
*"These biases reduced to 0.3 m (Mean offset for IfAG mainland mosaic for filtered, slopes less than 30 degrees), when only lower slopes are considered (Table 6). Furthermore, the accuracy of ICESat-2 is known to degrade at higher curvatures (Shen, X et al. 2022); the uncertainties of our DEMs may therefore be overestimated in such regions."*

References :

Howat, Ian, et al., 2022, 'The Reference Elevation Model of Antarctica - Mosaics, Version 2', https://doi.org/10.7910/DVN/EBW8UC, Harvard Dataverse. Accessed 2025-12-03
Howat, I. M., Porter, C., Smith, B. E., Noh, M.-J., and Morin, P.: The reference elevation model of Antarctica, The Cryosphere, 13, 665–674,395
2019.

https://www.agisoft.com/pdf/metashape-pro_2_2_en.pdf

https://portal.opentopography.org/datasetMetadata?otCollectionID=OT.082023.3031.1